# `IntroUNET`: Identifying introgressed alleles via semantic segmentation

**Dylan D. Ray[1], Lex Flagel[2,3], Daniel R. Schrider[1] ***

**1** Department of Genetics, University of North Carolina at Chapel Hill, Chapel Hill, North Carolina, United States of America, **2** Division of Data Science, Gencove Inc., New York, New York, United States of America, **3** Department of Plant and Microbial Biology, University of Minnesota, Saint Paul, Minnesota, United States of America

* drs@unc.edu

**Data Availability Statement:** All code required to train, test, and run the IntroUNET method are available on https://github.com/SchriderLab/introNets, as are links to the data analyzed in this study (all of which are publicly available).

## Abstract

A growing body of evidence suggests that gene flow between closely related species is a widespread phenomenon. Alleles that introgress from one species into a close relative are typically neutral or deleterious, but sometimes confer a significant fitness advantage. Given the potential relevance to speciation and adaptation, numerous methods have therefore been devised to identify regions of the genome that have experienced introgression. Recently, supervised machine learning approaches have been shown to be highly effective for detecting introgression. One especially promising approach is to treat population genetic inference as an image classification problem, and feed an image representation of a population genetic alignment as input to a deep neural network that distinguishes among evolutionary models (i.e. introgression or no introgression). However, if we wish to investigate the full extent and fitness effects of introgression, merely identifying genomic regions in a population genetic alignment that harbor introgressed loci is insufficient—ideally we would be able to infer precisely which individuals have introgressed material and at which positions in the genome. Here we adapt a deep learning algorithm for semantic segmentation, the task of correctly identifying the type of object to which each individual pixel in an image belongs, to the task of identifying introgressed alleles. Our trained neural network is thus able to infer, for each individual in a two-population alignment, which of those individual's alleles were introgressed from the other population. We use simulated data to show that this approach is highly accurate, and that it can be readily extended to identify alleles that are introgressed from an unsampled "ghost" population, performing comparably to a supervised learning method tailored specifically to that task. Finally, we apply this method to data from *Drosophila*, showing that it is able to accurately recover introgressed haplotypes from real data. This analysis reveals that introgressed alleles are typically confined to lower frequencies within genic regions, suggestive of purifying selection, but are found at much higher frequencies in a region previously shown to be affected by adaptive introgression. Our method's success in recovering introgressed haplotypes in challenging real-world scenarios underscores the utility of deep learning approaches for making richer evolutionary inferences from genomic data.

**Funding:** DRS was supported by the National Institutes of Health award number R35GM138286 and R01HG010774. DDR was supported by the Department of Genetics at the University of North Carolina at Chapel Hill and NIH award number R01AI153523. The funders had no role in study design, data collection and analysis, decision to publish, or preparation of the manuscript.

**Competing interests:** The authors have declared that no competing interests exist.

## Author summary

It is now known that a sizeable fraction of species occasionally hybridize with related species. Thus, many species harbor genetic material that traces its ancestry to closely related species. For example, many humans contain DNA that was "introgressed" from Neanderthals. The growing appreciation of the commonality of introgression has sparked a keen interest in determining which portions of the genome were introgressed. Several statistical approaches have been devised for identifying the population genetic signatures of introgression, but the most powerful techniques for this task take advantage of modern machine learning techniques. Here, we describe a deep learning method for identifying segments of introgressed DNA. This method is based on neural networks used to determine which pixels in an image belong to which type of object. By treating a matrix of genotypes from a sample of individuals from two closely related species, we can use this deep learning approach to accurately infer which portions of which genomes from the first population were introgressed from the second, and vice-versa. We show that our method, which we have released as an open-source software package, is highly accurate using a variety of simulated scenarios and a real test case from the genus Drosophila.

## 1 Introduction

Speciation events are often followed by the two nascent species coming into secondary contact. In many cases this creates the potential for hybridization, which can in turn result in alleles crossing from one species into the other [1]. There is a growing body of evidence such that post-speciation gene flow is a common occurrence [1–4]. The introgression of alleles from one species to another can have a significant impact on fitness and evolution. Introgressed alleles will presumably often reduce fitness in the recipient species, because of incompatibilities between the introgressed alleles and the recipient species' environment or genomic background [5, 6], or because the donor species may in some cases have a higher burden of deleterious alleles [7]. In rarer instances introgression may be beneficial, especially if the species have a shared selective environment, and the donor species contains alleles that that are adaptive in this environment and that the recipient species lacks (e.g. [8, 9]). For example, in humans, an *EPAS1* allele that originated in an archaic human relative (Denisovans) and that confers greater tolerance to high altitudes, is found at high frequency in Tibetans [10]. A similar observation of adaptive introgression at *EPAS1* was also made in Tibetan mastiffs, who may have received adaptive alleles from Tibetan gray wolves [11]. In *Anopheles* mosquitos, alleles that increase resistance to insecticides have jumped across species barriers [12]—an alarming observation that suggests that the control of these and other pests may be made even more challenging by their potential to experience adaptive introgression. These findings suggest that, while often deleterious, introgression may also present a route to more rapid adaptation in species that are able to borrow adaptive alleles from a neighboring relative.

For these reasons, there is a great deal of interest in detecting the extent of and genomic loci affected by introgression [13, 14]. A number of statistical tests have been developed to detect the presence of introgressed alleles/haplotypes. These may ask whether there is an excess of sites in the genome that appear to exhibit patterns of inheritance that depart from the known phylogenetic relationship between species [15–17] or an excess of phylogenetic trees inferred from individual loci that differ from the species tree in a manner that is best explained by introgression [3, 18, 19]. When genomic data from multiple individuals from a given

population are available, statistical tests may search for loci that have unusually similar allele frequencies between the populations experiencing gene flow [20–22], or even for haplotypes that appear to be shared between these populations [14, 23, 24]; the latter approach has the potential to identify specific loci affected by introgression. Local ancestry inference methods, which typically compare a sample of potentially admixed genomes to a reference panel of potential donor populations [25], also have the potential to reveal introgressed regions [26, 27].

Although methodological advances in the search for introgressed regions are welcome, merely assessing the presence of introgression within a genomic region has its limitations. We may wish to know how much introgression has occurred in a given region: how many sites were affected, and which individuals have introgressed material at each of these sites? Note that this information would in turn yield estimates of the frequencies of introgressed alleles in the recipient population. All of this information is useful for drawing inferences about the fitness effects of gene flow between a particular pair of populations, or even at particular loci. The development of machine learning methods for population genetic inference may represent one possible means of addressing this problem. Machine learning methods have recently made significant inroads in a number of problems in population genetics, including detecting positive selection [28–34], performing demographic inference [35, 36], and estimating recombination rates [37, 38]. We previously developed a supervised machine learning method for detecting gene flow, called FILET, that dramatically increases the power to detect introgressed loci relative to methods that use a single summary statistic [39]. More recently, Durvasula et al. created a machine learning method, called ArchIE, that infers, for each individual in a sample, whether they received introgressed alleles from an unsampled (or "ghost") population in a given window [40]. By averaging predictions made across all sliding windows overlapping a given polymorphism, ArchIE is capable of producing an inference at every polymorphism for each individual in the alignment.

Both FILET and ArchIE make their inferences by examining vectors of population genetic summary statistics and using simulations to train a classifier to distinguish among alternative evolutionary models, an approach that has become increasingly common in recent years [41]. However, an alternative approach that could potentially be even more powerful and flexible is to skip the step of calculating summary statistics and instead train deep neural networks to examine population genetic alignments directly as their input. For example, convolutional neural networks (CNNs; [42, 43]), which are powerful tools for making predictions from various data types including images [44], can readily be adapted to population genetic alignments as these can be treated as images, with the value at any given pixel indicating which allele/genotype a given individual has at a given cite. Chan et al. recently showed that this approach can detect recombination rate hotspots with excellent accuracy [45]. Flagel et al. showed that CNNs could be trained to solve a number of population genetic problems, including detecting selective sweeps and introgressed loci, and inferring recombination rates (see also [46]), with accuracy matching or exceeding that of previous state-of-the-art methods [47]. Subsequent studies have used CNNs to perform demographic inference [48], and detecting adaptive introgression [49]. A variant of a generative adversarial network, which seeks to distinguish between real and synthetic data, has also been used to estimate demographic parameters [50]. Additional population genetic tasks that artificial neural networks have been designed for include identifying the geographic location of origin of a genome [51], mapping genetic variation data to a low-dimensional latent space [52], and inferring dispersal distances in spatial populations [53].

Although the above examples all underscore the potential of deep learning for population genetics, for the most part they simply use classification (model selection) or regression

(parameter inference) to make a prediction for an entire region/genome. However, the extraordinary flexibility of deep learning architectures makes them suitable for problems that involve the production of far richer outputs. Indeed, deep learning has been used to generating artificial genomic data [54]. Another recent interesting example that treats genetic alignments as image data but produces more detailed outputs is Hamid et al's network, which localizes selective sweeps along a chromosome in admixed populations [55]. This method works by uses an object detection framework, which seeks to identify and draw bounding boxes around objects in an image—thus, an alignment is not only classified as having a sweep, but bounds are drawn around the location of the target of selection. An even more detailed form of image processing is semantic segmentation, where the goal is to produce a prediction for each pixel in an image identifying the type of object that this pixel belongs to. This is an ideal framework for detecting introgressed alleles, as we can in principle infer, for each allele in each individual (i.e. for each pixel), whether that allele was introgressed or not. Here, we describe IntroUNET, a fully convolutional neural network that examines a two-population alignment and seeks to identify which alleles in each individual were introgressed from the other population. We evaluate IntroUNET on simulated data where we show that it is able to infer introgression with high accuracy, including in scenarios of bidirectional gene flow. We also show that IntroUNET can be easily extended to detect ghost introgression. Finally, we examine the well-known case of introgression between *Drosophila simulans* and *Drosophila sechellia* [39, 56], demonstrating that IntroUNET can accurately identify introgressed alleles/haplotypes in challenging real-world scenarios.

## 2 Methods

### 2.1 Overview of method

In this paper we explore the potential efficacy of traditional fully convolutional neural networks (FNNs) to detect introgressed alleles in population genetic alignments, provided that the user can supply a set of training examples where the precise introgressed haplotypes, and the individuals that harbor them, are known. Typically these training examples will be simulated under a demographic model (or a set of likely demographic models) that have been estimated from the population(s) under study. FNNs were first designed to tackle image-to-image semantic segmentation, wherein a prediction is made for each pixel in an image. Specifically, these networks take multi-channel images (usually with three color channels for red-green-blue (RGB) or hue-saturation-value (HSV)) as input and return an output of the same width and height with the output pixels denoting the type of object to which the pixel belongs [57]. Here, we adapt a FNN for semantic segmentation to classify each allele/genotype for each individual in a population genetic alignment as introgressed or not. Our approach is to treat a multi-population alignment of a genomic window as a tensor whose dimensions are $l \times n \times m$, where $l$ is the number of populations in the sample, $n$ is the number of (haploid) genomes in each population, and $m$ is the number of biallelic segregating sites in the genomic window. Note that $l = 2$ in all experiments in this paper; one could in principle adapt this approach to examine $l > 2$ populations by simply adding additional input channels, but one would also have to decide how to order these population samples in the input (see below) and we caution that we have not evaluated whether acceptable accuracy can be obtained with higher values of $l$. The value for each entry in the tensor is a binary indicator specifying whether the derived allele (or minor allele, in unpolarized data) is present in a given genome at a given polymorphism in a given population. On the basis of this input tensor, the FNN infers another $l \times n \times m$ binary tensor that specifies which alleles were introgressed from the other population and which were not, thus framing the problem as image-to-image segmentation. We note that

IntroUNET does not include information about monomorphic sites or the physical distances between segregating sites in the alignment.

**2.1.1 Ordering individuals within the input image.**   Unlike the rows of pixels in an image, the ordering of rows in population genetic alignments is not typically meaningful. Thus, for the same set of population genomic sequences, there are $n!$ possible image representations we could produce that all have the exact same information, meaning that the function that we would like our neural network to learn is far from isomorphic (a one-to-one mapping). One approach to deal with this problem is to use an exchangeable neural network (following [45] and [58]) where only permutation-invariant functions (column-wise means/maxima, etc.) are applied along the "genomes" axis of the input thereby ensuring that the ordering of rows has no bearing on the output. However, standard FNNs, which were inspired by the arrangement of cortical neurons in mammalian eyes [59], use 2D convolutions which rely on the hierarchical and spatially local information inherent in visual images to learn relevant "filters" or kernels, and are by definition non-exchangeable.

One way to potentially induce hierarchical information in population genetic data, while simultaneously mitigating the many-to-one mapping issue, is to sort the individual genomic windows in a manner that may be meaningful in the context of the regression model being attempted. However, there is no obvious choice of ordering that would be the most meaningful. One could imagine using the topological order specified by the genealogical trees that the alignment resulted from, but these are not known in real data and must be inferred, and in recombining genomes the tree topology and branch lengths will vary along the sequence [60]. Flagel et al. [47] addressed this problem by sorting individuals by sequence similarity by constructing a nearest neighbor graph with some distance metric, finding that this yielded an appreciable boost in accuracy relative to unsorted alignments. It is this approach that we build on here.

One way to induce a visually meaningful structure in randomly ordered N-dimensional vectors is to use seriation. Seriation, otherwise referred to as ordination, is a statistical method which seeks the best enumeration order of a set of described objects [61]. More precisely, the method seeks to order vectors in such a way that the distance between neighboring vectors in the sequence is small and thus the total summed distance over neighboring pairs is minimized. The choice of distance metrics includes any metric function defined for multi-dimensional real fields; examples include Euclidean distance and Manhattan distance for binary vectors. Such a sorting would be similar to ordering individuals by topological order along the "average" tree describing their relatedness within the focal genomic region. As a consequence, if multiple individuals had introgressed alleles, this approach would create blocks of introgression that may be more conspicuous and thus easier for the neural network to detect. We experimented with different distance metrics as described in the Results.

Seriation is known to be NP-hard, i.e. it is intractable to find exact or globally optimal solutions because as the number of vectors or destinations, $n$, grows large, the complexity grows faster than any finite polynomial function of $n$ [62]. However, when the distance measure between vectors is defined to be a metric, as we do here, seriation has been found to be APX-complete or approximable-complete, i.e. there exists algorithms which can approximate the global optimum in polynomial time given some asymptotic error bound [62]. For this project we use Google's OR-tools (https://developers.google.com/optimization/), which performs well even on large samples.

Seriation orders a single population of vectors, but in our data schema there are two populations in the input tensor (dimension 1), and thus any given row in the image corresponds to two haploid individuals: one in population 1, and the other in population 2. The correspondence between these two overlying individuals may impact the ability of the neural network's

training algorithm to find meaningful 2D convolution filters. We addressed this problem by seriating one population, and then ordering the second population by performing least-cost linear matching between the seriated population and the other. Linear matching was performed between populations using the scipy Python package's functionality which employs the Kuhn-Munkres algorithm, also known as the Hungarian method, to do so [63, 64]. Fig 1 shows, for a simulated two-population alignment, an example image representation of the input and output as specified in this section. We recommend selecting the more diverse of the two populations (as measured by nucleotide diversity [65, 66]) first, and then matching the less diverse population to the first. If the two populations have very similar levels of diversity, then this choice is unlikely to be consequential.

To summarize the intuition for the ordering and matching of populations in the image: convolutional features were designed to capture hierarchical, spatially localized information, such as stretches of similarity between individuals of different populations, and seriation and least-cost matching procedures are done to make these similarities easier to detect by the convolutional filters. The effect of these procedures on the input representation can be seen in Fig 1: note that in this example, the introgressed individuals in population 2 tend to be matched in their location along the $y$-axis with similar individuals from population 1.

**2.1.2 Overview of neural network optimization.** Here we briefly describe some terminology and methods common to machine learning. Neural network inference is a type of non-linear regression and the task of "training" a neural network, i.e. finding optimal parameters that make up the non-linear regression, is often broken up into two categories, supervised learning and unsupervised learning. Supervised learning uses labeled input and output data and so the ground truth labels or desired values for $y$ are known in advance, whereas unsupervised learning does not include these. For the entirety of this paper we use a supervised approach where our ground truth binary labels are obtained via population genetic simulation.

In supervised learning for neural networks, an optimal parameter set, or one that correctly predicts the known $y$ variables given $x$ is often sought through a process called mini-batch gradient descent sometimes called stochastic gradient descent [67]. Given $\theta$, the learnable parameters in our neural network architecture, we first define some objective function $\mathcal{L}(\theta, \hat{y}, y)$ which we seek to minimize and common choices include the mean squared error between scalar predictions, or binary cross entropy (defined below) for classification problems just as they are used to define scalar regression and logistic regression respectively. In gradient descent we update the weights, $\theta$, by computing the gradient of $\mathcal{L}$ with respect to $\theta$ and moving in the opposite direction:

$$\theta := \theta - \eta \nabla \mathcal{L}(\theta, \hat{y}, y) \tag{1}$$

where the gradient $\nabla \mathcal{L}$ is the vector of partial derivatives of $\mathcal{L}$ with respect to each scalar weight $\theta_i$, and $\eta$ is small quantity called the learning rate that sets the size of the step taken each iteration. In practice however, the loss function cannot be computed over all known examples in the labeled dataset due to the limited computer memory, the large number of often high-dimensional examples, and the number of weights involved. Thus, mini-batch gradient descent estimates the gradients expected value via a sample of the dataset called a batch each step:

$$w := w - \eta \nabla \mathcal{L}(\theta, \hat{y}, y) = \theta - \frac{\eta}{N} \sum_{i=1}^{N} \nabla \mathcal{L}_i(\theta, f(\theta, x_i), y_i) \tag{2}$$

where $\mathcal{L}_i(\theta, f(\theta, x_i), y_i)$ represents the loss function computed on a single example and $N$ represents the batch size or number of examples in each sample.

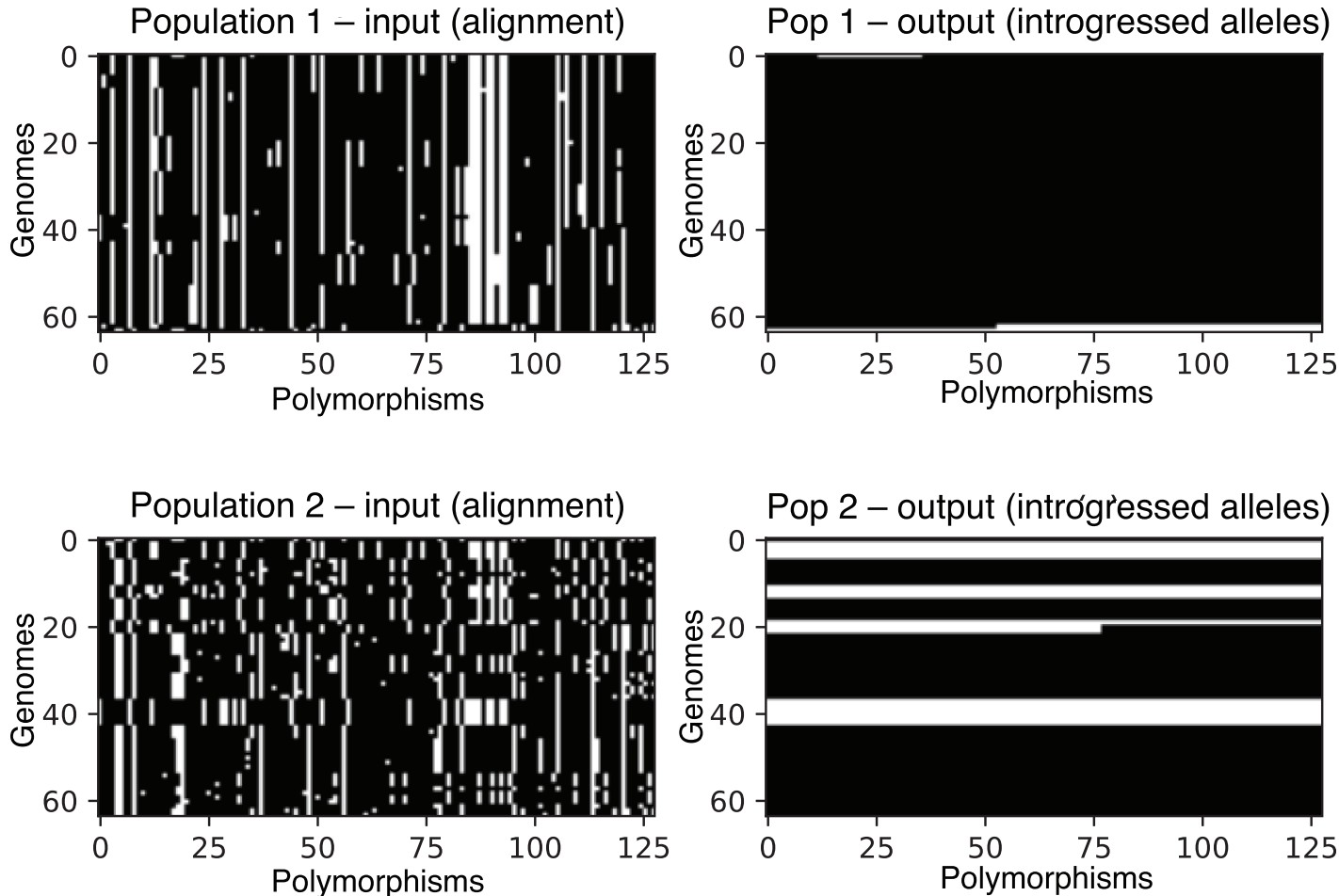

**Fig 1. Image representation of an example input tensor (left column) and its corresponding output (right column), from a simulated scenario of bidirectional gene flow.** Here, the two populations are shown as separate matrices, although they are actually part of the same input tensor (i.e. they are the two values along the "channel" dimension in the tensor). The input alignments are represented as black and white images where the ancestral allele is shown in black and the derived allele in white. The output matrices show the locations of alleles in a recipient population that were introgressed from the donor population. Thus, the white pixels in the output for population 1 show alleles that were introgressed from population 2, and the white pixels in the output for population 2 represent alleles introgressed from population 1.

**2.1.3 Network architecture, training, and evaluation.** We chose to use a variant of the UNet++ architecture [68], a type of U-Net [69]. U-Nets are fully convolutional neural networks that have proved capable of achieving excellent performance in semantic segmentation problems, and are so-named because of the shape of their architecture (e.g. Fig 2). U-Net++ is a slightly modified version of the original U-Net that was proposed by Zhou et al. [68], who showed that UNet++ outperforms its predecessor in medical image segmentation. UNet++ shares the same "backbone" of convolution, upsampling, and downsampling operations as the original U-Net, but with added nested skip connections like those shown in Fig 2 that aim to reduce the loss of information between the encoder (i.e. the left side of the "U") and decoder (the right side of the "U") sub-networks. Our network architecture is a variant of U-Net++ in that we also use the nested skip connections, but the sizes of our upsampling, downsampling, and convolution layers differ from been altered from [68] to suit the image sizes used here.

Fig 2 shows an outline of the architecture we chose. In the left side of the "U" a traditional CNN is trained to extract relevant features that become more and more coarse-grained due to

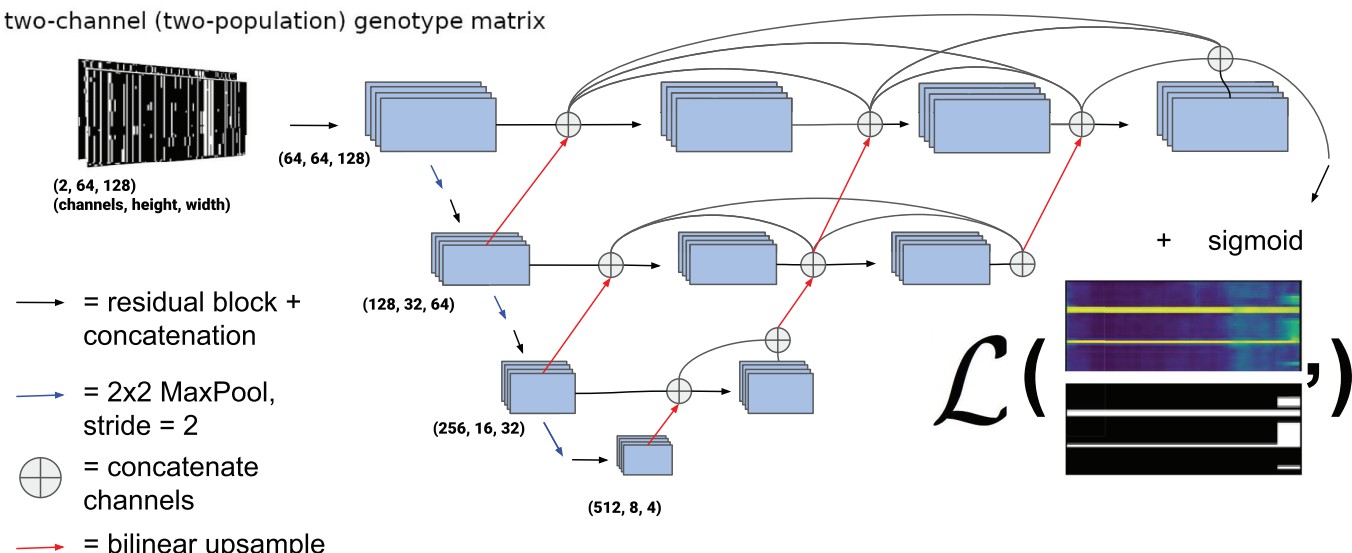

**Fig 2. UNet++ type architecture [68] used for all the problems in this paper.** The black arrows represent a residual block consisting of two convolutions where each convolution in the series is summed to the previous, and the convolution layers are concatenated before a non-linear activation (ELU) [74] is applied. The example output of the network is color scaled from 0 to 1 and represents the probability of introgression at a given allele for a given individual. The loss function (represented by the bold $\mathcal{L}$) is computed with the ground truth from the simulation and is the weighted binary cross entropy function (Eq 3). The weights and biases of the convolution operations are updated via gradient descent during training. The architecture we use for the problems discussed actually contains four down and up-sampling operations rather than the three portrayed here.

downsampling, and then the information is passed via upsampling and concatenation to the right side of the "U" finally resulting in a binary segmentation, or binary classification for each allele in each individual (in our case as introgressed or not introgressed). In this version of a U-Net, fine-grained and coarse-grained information are synthesized together using upsampling and skip connections. Note that in this network, upsampling steps refer to 2-dimensional bilinear upsampling, rather than transposed convolution. Each residual block consists of two convolution layers where the last layer is summed with the output of the first, after which these two outputs are concatenated. S1 Fig shows a diagram of our chosen residual block architecture. The kernel size (i.e. the size of the filters whose weights are learned during training) of each convolution layer is 3x3 pixels. The number of these filters (or feature maps) in the convolution operators for the decoder and encoder portions of the network was chosen to be increasing integer powers of 2 beginning with 32 (after the concatenation in residual blocks) with the largest number of filters in a given layer being 512. The depth of the network (i.e. the number of downsampling/upsampling operations) was chosen to be 4.

Although we did not conduct systematic hyperparameter optimization for this architecture, we did experiment some with different hyperparameter values and in general found that large networks (i.e. networks with more layers or more learnable parameters) performed better than smaller and more shallow networks. However, in general we found that, for this particular network architecture, the choice of hyper-parameters had a limited impact on performance. We therefore applied this architecture to each problem considered in the paper. We implemented this network, in PyTorch [70]. For regularization, we employed both 2D InstanceNorm [71] and dropout with a rate of 0.1 in each residual block of the network. We've made the code publicly available and open-source on GitHub under a GNUv3 license (https://github.com/SchriderLab/introNets).

We chose to using label smoothing in addition to the other forms of regularization present (i.e. batch normalization and dropout). Label smoothing regularization has been shown

empirically and theoretically to reduce the variance of the noise in estimating the gradient for optimization problems when using stochastic gradient descent [72]. Label smoothing randomly introduces uncertainty in the labels of batches via:

$$y^* = y(1 - \epsilon) + \frac{\epsilon}{2},$$

$$\text{where } \epsilon \sim U(0, \alpha)$$

Thus, true labels of 0 are perturbed up and labels of 1 are perturbed down via a randomly uniform distribution with a max of $\alpha$ which can be thought of as the strength of smoothing. For this problem, we experimented with three different values of $\alpha$: 0.001, 0.01, and 0.1. We found that a smoothing strength of $\alpha = 0.01$ yielded the best validation accuracy (S2 Fig).

For network training, we used the Adam optimizer with default settings [73]: a learning rate of 0.001 and $\beta_1, \beta_2 = 0.9, 0.999$. For all problems considered we trained the network for a maximum of 100 epochs, with a patience of 10 epochs (meaning we halted training if for ten consecutive epochs validation failed to decrease below the previous minimum). For our loss function, we used the weighted binary cross-entropy function:

$$\mathcal{L}(\theta) = \frac{1}{N} \sum_{n=1}^{N} H(p_n, q_n) = -\frac{1}{N} \sum_{n=1}^{N} \left[ w_p y_n \log \hat{y}_n + (1 - y_n) \log(1 - \hat{y}_n) \right] \tag{3}$$

where $H(p_n, q_n)$ is the cross-entropy function of the true and predicted label distributions ($p$ and $q$. respectively), the weight $w_p$ is the weight for positive (introgressed) examples and was set to the ratio of negative (non-introgressed) examples to positive examples, $y_n$ and $\hat{y}_n$ are the true and predicted labels for example pixel $n$, respectively, and $N$ is the total number of example pixels in the training set. Note that our neural network used a softmax activation function for the output layer, meaning that in the loss function above the prediction $\hat{y}_n$ is a continuous value ranging from 0 to 1. We use the weighted form of cross-entropy because the datasets examined in this paper are unbalanced in that there are far more non-introgressed alleles across individuals than introgressed alleles, and this may impact performance as the network will learn to place less value on the rarer class. Class weights are a commonly used approach for mitigating the effects of data imbalance during training [75, 76], and weighting by the relative frequency of each class as we have done here is one popular approach for setting these weights [77]. Note that this data imbalance can affect evaluation metrics such as the ROC curve, which plots a method's true positive rate against its false positive rate, thereby showing how much specificity must be relaxed (by decreasing the detection threshold) to achieve a given sensitivity. For this reason we also compute precision-recall (which shows how a method's positive predictive value decreases as its sensitivity increases) which is known to be robust to unbalanced binary inference problems as it does not incorporate correctly predicted absences [78]. We also note that class-imbalance does not bias our confusion matrices obtained from these evaluation sets, and we also report unbalanced classification accuracies obtained by simply averaging the percentages along the main diagonal of the confusion matrix.

It is worth mentioning that our network has 4 pooling operations (and, in the second half of the network, four upsampling operations) each of which leads to a reduction (and increase in the second half of the network) in both the width and height of the input by a factor of 2. Thus, both the width and height of the input image (individual and SNP axes) must be multiples $2^4 = 16$. In the problems considered in this paper, we chose to upsample the number of individuals in the dataset to the nearest multiple of 16, and always choose SNP-window sizes which are multiples of 16. When applied to real datasets, our tool upsamples the number of

individuals as necessary, which will result in multiple predictions made for some individuals, in which case our tool arbitrarily takes one of these predictions to be the final prediction for that individual. This may somewhat decrease the variance in predictions obtained for those individuals arbitrarily chosen to be upsampled, but this does not appear to have a major detrimental impact on accuracy. In the following sections, problem-specific details of batch size and other training specifications are given when needed. Finally, we note that in all analyses in this paper, if the $\hat{y}$ predicted for an allele by IntroUNET was greater than 0.5, then we interpreted that as a positive prediction (i.e. the allele is inferred to be introgressed), and otherwise as a negative prediction (no introgression).

**2.1.4 Window-based classifier for detecting introgression.**   Previously developed machine learning methods have been shown to be highly effective at identifying genomic windows that have experienced introgression [39, 47]. As Discussed below, we found that when applying IntroUNET to a dataset of *D. simulans* and *D. sechellia* genomes, accuracy was best when we examined windows previously shown to have evidence for introgression (using results from [39]). Because users may wish to incorporate such a step into their analyses of introgression, we decided to incorporate a classifier into the IntroUNET software package that, similar to that of Flagel et al. [47], examines an image representation of population genetic alignment and classifies a region as having experience introgression from population 1 to population 2, introgression from population 2 to 1, or no introgression. Specifically, we trained a discriminator with a ResNet34 architecture [79], a CNN that performs competitively on image classification tasks. Other than the number of input channels (changed from three to two in this case), we did not modify the architecture from its original implementation. We demonstrated the accuracy of this classifier by training and testing it using the same simulated dataset that was used to train IntroUNET to identify introgressed haplotypes in *D. simulans* and *D. sechellia*. This dataset was then filtered for simulations that didn't contain the desired introgression case, class-balanced via downsampling, and then split into a training and validation sets with the validation set comprising 5 percent of the total number of simulations for each case. We used the categorical cross entropy function as loss and the Adam optimizer [73] with the same settings as for the IntroUNET segmenter described above. The network was trained for a maximum of 100 epochs with a patience of 10 epochs.

## 2.2 Simulated introgression scenarios

**2.2.1 A simple simulated test case.**   We assessed our method's performance on a simple simulated scenario where two subpopulations, each consisting of $N = 500$ diploid individuals, split $4N$ generations ago and later experienced a pulse of gene flow. The time of the introgression event, and the fraction of individuals introgressed, was allowed to vary uniformly from replicate to replicate. The full list of parameters and values for this model is shown in Table 1. Note that we simulated fairly small population sizes here for computational tractability, and therefore used large mutation and recombination rates, $\mu$ and $r$, respectively, such that $4N\mu = 4Nr = 0.02$.

We simulated populations for this scenario using the evolutionary simulation software SLiM [80]. We simulate equal numbers of replicates of three scenarios: unidirectional introgression from population 1 to population 2, unidirectional introgression from population 2 to population 1, and bidirectional introgression. For each case, $10^5$ replicate 10 kb regions were simulated and a predictor and target image was created for each as described above. For the bidirectional case the target variable has two channels: one corresponding to whether a given allele was introgressed from population 1 to population 2, and the other corresponding to whether the allele was introgressed from population 2 to population 1. In the two

**Table 1. Parameters of the simple simulated test case.** We begin with a single population of size $N$ which is allowed to "burn in" for $20N$ generations so that the populations reach, or at least approach, equilibrium. Then, a split occurs $t_S$ generations ago. Next, after some amount of time of complete isolation, which follows the described uniform distribution, a pulse migration event occurs with individuals migrating with a probability also drawn from a uniform distribution. This migration event can occur in either direction or in both directions, and both unidirectional and bidirectional introgression is examined the Results. Note that in the case of bidirectional migration a separate rate is drawn for both directions, and the maximum value of this rate is one half that for unidirectional migration. Migration rates specify backward probabilities (i.e. the expected fraction of the recipient population that migrates from the source population during the introgression event).

| Parameter | Value |
|---|---|
| Simulated chromosome size, $L$ | 10 kb |
| Sub-population size, $N$ | 500 |
| Burn-in time (generations) | $20N$ |
| Split time (generations ago), $t_S$ | $4N$ |
| Time of introgression event (generations ago) | $U(0, 0.25t_S)$ |
| Unidirectional introgression probability per individual | $U(0.1, 0.5)$ |
| Bidirectional introgression probability per individual (in each population) | $U(0.1, 0.25)$ |
| Mutation rate | $1.0 \times 10^{-5}$ |
| Recombination rate | $1.0 \times 10^{-5}$ |
| Sample size, population 1, $n_1$ | 32 |
| Sample size, population 2, $n_2$ | 32 |

unidirectional cases there is only a single output channel denoting whether or not there was introgression in the direction being considered. Note that we did not explicitly simulate examples where migration was disallowed, but in most simulated examples the majority of pixels in the alignment image were not affected by introgression.

Our initial set of $10^5$ replicates was split into training and validation sets (95% training, 5% validation), and an additional evaluation set of 1000 replicates was simulated separately but with the same parameter distribution as the training and validation sets. For this experiment we excluded from both training and testing the small number of simulation replicates that by chance had no introgressed alleles in the sampled individuals. We used a batch size of 16 examples and trained for one hundred epochs or until experiencing ten consecutive epochs with validation loss failing to achieve a new minimum (i.e. patience = 10). The model with the lowest validation loss obtained was then used on the evaluation to obtain the metrics reported. An example input along with true and inferred outputs of the bidirectional scenario is shown in Fig 3A.

We also generated a test set incorporating background selection into this scenario of introgression. To accomplish this, we used `stdpopsim` version 0.2.0 [81, 82] to generate `SLiM` scripts simulating negative and background selection using a genetic map [83] and distribution of fitness effects [84] for mutations in exonic regions, and exon annotations (the `FlyBase_BDGP6.32.51_exons` set in `stdpopsim`, taken from FlyBase [85]) all obtained from *D. melanogaster*. We then programmatically modified the `SLiM` scripts to include bidirectional introgression under the same scenario examined above, which each script generating one test replicate of a 1 Mb region with recombination and annotation data taken from chr3L, before running `IntroUNET` on the central 100 kb of each test example.

**2.2.2 Training a U-Net to detect ghost introgression.** We sought to assess the effectiveness of our approach on the problem of detecting ghost introgression. Specifically, we followed the scenario of Durvasula et al. [40], where the goal is to detect introgression from an unsampled population when given data from the recipient population and an un-introgressed

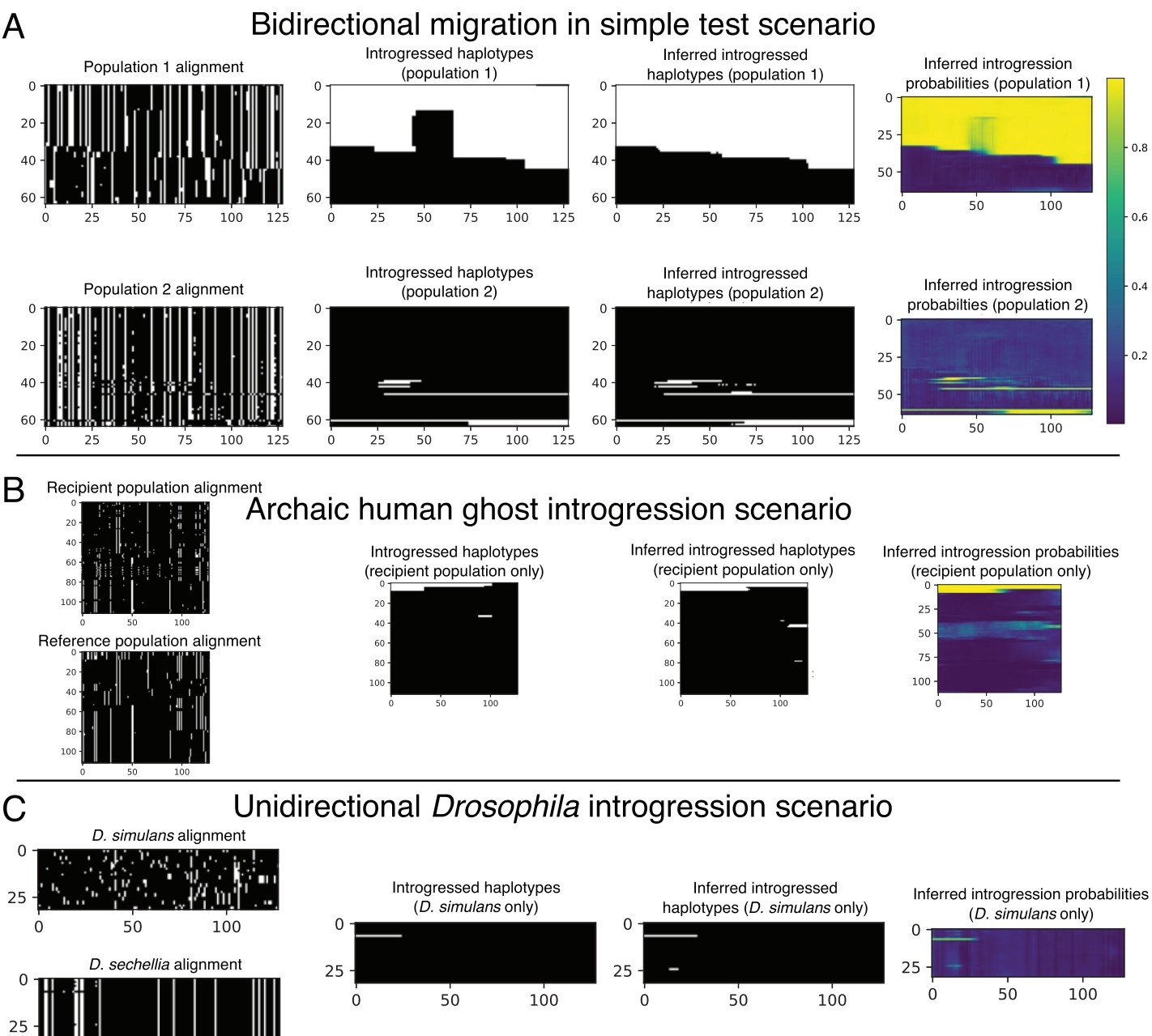

**Fig 3. Example inputs and outputs (both true and inferred) for each of the three problems we used to assess `IntroUNET`'s effectiveness.** (A) A simulated example of the simple test scenario of a two-population split followed by recent single-pulse introgression event (bidirectional, in this case). The first column shows the population genetic alignments for this example, with the two panels corresponding to the two input channels (population 1 and population 2). The second shows the true histories of introgression for this example (again, with white pixels representing introgressed alleles); note that both population 1 and population 2 have introgressed alleles. The third and fourth columns show `IntroUNET`'s inference on this simulation, with the former showing the most probable class (i.e. introgression or no introgression) for each individual at each polymorphism, and the latter showing the inferred probability of introgression (i.e. the raw softmax output for the introgression class). The color bar for these plots is shown in panel (A), and the scaling is the same for the panels below as well. (B) A simulated example of the archaic ghost introgression scenario. The four columns are the same as in panel (A), but here we are examining a recipient population and a reference population, with the goal of identifying introgression only in the former. Thus, our output has only one population/channel. (C) A simulated example of our *Drosophila* introgression scenario. The four columns are the same as in (A) and (B), and here we are concerned with identifying introgression from *D. simulans* to *D. sechellia*, so again our output has only one channel (i.e. introgressed alleles in *D. sechellia*).

"reference" population. We used the same neural network architecture described above for this problem, but note that the two input channels are the recipient and reference populations, and there is only one output channel, indicating whether or not a given allele in the recipient population was introgressed from the donor ghost population. Here we train under the introgression scenario from [40], which involves a split of the ghost population, followed by the split of the reference and the recipient populations, and later followed by introgression from the ghost population to the recipient population. A diagram of this model is shown in S12 Fig, and the parameters used to simulate it are shown in Table 2. Simulated alignments were generated using `msmodified`, a version of Hudson's coalescent simulator `ms` [86] modified by Durvasula et al. to track introgressed alleles in the presence of unidirectional gene flow. This was done using the simulation command line from the GitHub repository associated with ref. [40] (https://github.com/sriramlab/ArchIE/).

We simulated $10^6$ training/validation replicates with 5 percent of the data being randomly chosen for validation, using a window size of 50 kb and filtering windows that had no introgression. We chose to filter examples with no introgression to slightly upsample the amount of introgressed alleles as the problem is heavily imbalanced. We used the training set obtained to estimate the ratio of positive to negative alleles to weight the loss function as we describe in Eq 3. We simulated 1000 replicates separately with a window size of 1 Mb to evaluate both methods on. For this problem, we used an input size of $2 \times 112 \times 192$, corresponding to 2 populations (the recipient and the reference populations), 112 individuals in each population, and 192 polymorphisms in each window examined, again with each entry being 0 (the ancestral allele) or 1 (the derived allele). The original simulation command given by [40] gave 100 individuals per sub-population, and our images of 112 individuals (the nearest multiple of 16) were created via up-sampling (i.e. arbitrarily selecting 12 individuals to duplicate in the input tensor). Our target image is simply the alleles that were introgressed from the archaic population into the recipient population represented in the form of a $192 \times 112$ binary matrix. We used a batch size of 32 when training the neural network for this problem. An example input along with true and inferred outputs of this scenario is shown in Fig 3B.

We compared the performance of our method on this task to `ArchIE`, the logistic model that Durvasula et al. created to solve this problem [40]. Briefly, `ArchIE` uses a feature vector generated for each (haploid) individual in the reference population to predict whether the individual contains introgressed alleles in a given window. `ArchIE` then obtains predictions for individual polymorphisms by using a sliding window and averaging the predictions for the focal individual across all sliding windows overlapping the focal site. The features used by

**Table 2. Parameters of the ghost-introgression demographic model, reproduced from Table. 1 from Durvasula et al. [40].** Note that our simulations used the command from Durvasula et al.'s GitHub repository (https://github.com/sriramlab/ArchIE/, which also contains a brief bottleneck experienced by the ghost population).

| Parameter | Value |
|---|---|
| Reference population size, $N_1$ | 10000 |
| Target population size, $N_2$ | 10000 |
| Archaic population size, $N_a$ | 10000 |
| Archaic split time (generations ago) | $1.2N$ |
| Split time (generations ago), $t_S$ | $0.25N$ |
| Time of introgression event (generations ago) | $0.2N$ |
| Fraction of individuals introgressed | 0.02 |
| Mutation rate | $1.25 \times 10^{-8}$ |
| Recombination rate | $1.0 \times 10^{-8}$ |

ArchIE to make a classification for each focal individual include: the individual frequency spectrum (IFS), which is a vector showing the number of mutations present on an individual haplotype that are found at a given derived allele frequency within the recipient sample; a vector containing the Euclidean distance between the focal individual and each other individual in the recipient sample, as well as the mean, variance, skewness, and kurtosis of the distribution of these distances; the minimum distance between the focal individual and all the individuals within the reference sample, the number of singletons found in the focal individual (i.e. the derived allele is absent from both all other individuals in both samples); finally, the S* statistic [87] is included in the vector. When training ArchIE using the same training data used for IntroUNET on this problem, we found that we did not achieve accuracy comparable to the original publication unless we balanced the training set. We accomplished this by randomly downsampling the number of training vectors in the non-introgressed class until the total number of vectors in each class was equal (resulting in a total training set size of $\sim 2.6$ million vectors combined across both classes). Note that no balancing was done when training IntroUNET.

For the sake of a fair comparison, the evaluation set of 1000 was kept the same between the two methods. We note that, unlike ArchIE, IntroUNET's window size and stride are specified in polymporphisms rather than base-pairs and were chosen to be 192 and 16 polymorphisms respectively; when averaging predictions across windows, ArchIE used a window size of 50 kb and a step size of 10 kb. We also used a Gaussian window to weight predictions close the edges of the "image" smaller when averaging. This choice was made to mitigate potentially poor predictions close the edges which is known to be an issue for architectures that employ 2-d convolution due to the necessary padding operation that takes place in this part of the input tensor.

**2.2.3 Application to real data: Finding introgressed regions in _D. simulans_ and _D. sechellia_.** To assess our method's practical utility, we applied it do the dataset of 20 _D. simulans_ inbred lines [88] and 14 wild-caught _D. sechellia_ genomes (i.e. 7 phased diploids) previously examined in [39]. First, we obtained genotypes, phased haplotypes, and trained our method from simulated data in a manner similar to that described in [39]. Following [89], we used ∂a∂i to estimate the parameters of a two-population isolation-with-migration demographic model allowing for exponential population size change following the population split. This was done after mapping reads, calling variants, and phasing haplotypes (via shapeit2 [90]) in the same manner as described previously [39]. In this instance, we mapped reads to FlyBase's [91] release 2.02 of the _D. simulans_ reference genome [92] using BWA version 0.7.15 [93].

When running ∂a∂i, we used the same optimization procedure as described previously [39], and once again calculated the SFS only using intergenic polymorphisms located at least 5 kb away from the nearest protein-coding gene. In our previous analysis, we had accounted for uncertainty in our estimation of demographic parameters by drawing each parameter from an arbitrarily chosen uniform distribution centered around the parameter point estimate [39]. For this study, we instead ran ∂a∂i in a bootstrapped fashion by selecting with replacement which of the contiguous intergenic regions at least 5 kb from genes would be included (i.e. the allele frequencies at polymorphisms in these regions would be included in the joint-SFS for the bootstrap replicate). This was repeated 100 times, and for each bootstrap replicate, we used the same optimization procedure as described previously [39] to obtain 10 separate demographic parameter estimates (each beginning from a different randomly chosen point in parameter space) for each bootstrap replicate. Only replicates with at least 5 successful optimization runs were retained. Of the 57 bootstrap replicates that satisfied this criterion, we recorded the

maximum log-likelihood obtained for each replicate along with corresponding parameter estimates.

In section 4, we list the demographic parameters and their estimated log-likelihood for each bootstrap replicate run of $\partial a \partial i$. When examining these results we noticed that a 14 replicates yielded maximum log-likelihood scores that were substantially lower than those of the remaining replicates (i.e. $<-2500$ while all other scores were $>-1750$), and we opted to retain only those 43 replicates for which this log-likelihood value was greater than $-1750$. For each of these parameter sets we generated 5000 simulation replicates of 10 kb regions using `msmodified`, resulting in 215,000 replicates total. For these simulations, the mutation and recombination rates were set to $5 \times 10 - 9$ per bp. Note that although we did allow $\partial a \partial i$ to infer continuous migration rates, we did not include these in our training/test simulations. Instead, we used `msmodified` to cause a pulse-migration event from *D. simulans* to *D. sechellia* to occur in each replicate, at a time drawn uniformly from between 0 and $0.25 \times N_a$ generations ago, where $N_a$ is the ancestral population size. The probability that any given individual in the *D. sechellia* population descended from a migrant was drawn uniformly from 0 and 1.0. We again limited the U-Net training and validation set to only those windows which contained introgressed alleles, which left 188,644 replicates total to be split into training and validation, using a random five percent of the data for validation. A separately simulated test set was generated in the same fashion, consisting of 1000 replicates containing introgressed alleles; this set was used to produce evaluation metrics reported in the Results. Code for generating simulations from the bootstrap replicates shown in section 4, can be found at https://github.com/SchriderLab/introNets/blob/main/src/data/simulate_msmodified.py.

Neural networks are often found to be poorly calibrated or biased in their predictions resulting in overconfidence or underconfidence in the posterior probabilities, even if they achieve high classification accuracy [94]. Platt scaling simply finds a line of best fit (a slope and intercept) for the predicted logistic scores given by a pre-trained network and the ground truth via the unweighted cross entropy function. We observed poor calibration in our test simulations for this *Drosophila* model. Thus, After training, we then learned a posterior probability correction via Platt scaling [95], which we found to produce better calibrated estimates of the probability of introgression than the raw output from `IntroUNET`. The recalibrator was trained via gradient descent for 25 epochs on the unweighted cross entropy loss calculated on the validation set. This was accomplished via PyTorch and the code to do so is also included in the `IntroUNET` GitHub repository. We note that Platt scaling was used in both on our simulated evaluation set and when applying it to real data below, giving us better confidence in the posterior probabilities computed in our analysis of predicted introgressed haplotypes in *D. sechellia*. An example input along with true and inferred outputs of the *Drosophila* simulated scenario is shown in Fig 3C. We did not compute Platt corrections for the other problems examined in this paper, as we did not examine posterior probabilities for further analysis, and classification accuracy was adequate without this scaling. We also note that ROC and precision-recall curves are not expected to be impacted by recalibration.

As described in the Results below, the *Drosophila* version of our U-Net performed best when some introgression was present in the input alignment. We therefore focused our analysis on regions that we previously found to contain introgression in the direction of *D. simulans* to *D. sechellia* (data obtained from https://github.com/kr-colab/FILET/blob/master/simSechResults/intro_regions_sim_to_sech_clustered_flybase2.02.bed), and examined our results with respect to version 2.02 of the FlyBase annotation [91] for the *D. simulans* genome. We note that this iteration of the `IntroUNET` occasionally produced false positive predictions where all of the *D. sechellia* genomes were inferred to be introgressed—we speculate that this is due to the low degree of diversity within the *D. sechellia* population resulting in false positive

introgression calls being repeated across individuals. Thus, to reduce the false positive rate in our analysis of the real dataset, we ignored sites that were predicted to be introgressed in more than half of our *D. sechellia* genomes. At sites that were inferred to have experienced introgression, we recorded the fraction of *D. sechellia* individuals inferred to be introgressed an introgressed allele at that site, and used this as our estimate of the frequency of introgressed haplotypes at that site.

## 3 Results

In the following section we evaluate the performance of IntroUNET on simulated data on three different scenarios (see Fig 3 for example inputs/outputs for each). The third of these scenarios is the case of introgression between *D. simulans* and *D. sechellia* [56] for which we also have a two-population sample [39] that we then use to demonstrate method's performance on real data. We then examine two practical considerations for our method: the effect sorting of individuals within an alignment, and IntroUNET's computational cost, respectively.

### 3.1 IntroUNET accurately identifies introgressed alleles in a simulated dataset

After designing our IntroUNET as described in the Methods, we sought to assess its effectiveness on simulated data. We began with a simple two-population model with constant population sizes, a split time of $4N$ generations ago, and introgression events occurring at times ranging between 0 and $N$ generations ago (see Methods and Table 1 for more detail and Fig 3A for example input and output for this problem). We evaluated IntroUNET's accuracy under three scenarios: introgression from population 1 to population 2, from population 2 to 1, and bidirectional introgression between both populations. We find that accuracy is very high in both unidirectional cases (e.g. area under ROC curve, or AUC, $\sim 0.99$, and area under precision-recall curve, or AUPR, $\sim 0.98$; with ROC curves, precision-recall curves, and confusion matrices shown in Fig 4A and 4B). The examination of the two unidirectional cases is a useful sanity check, as the two populations experience identical demographic histories and thus performance should be similar in both cases, which is indeed the case. Accuracy is slightly reduced in the bidirectional case (AUC $\sim 0.98$ and AUPR $\sim 0.93$; Fig 4C), which may be expected as this is a more difficult problem because individuals in either population may trace their ancestry to the other, perhaps making inter-population comparisons for the UNET more difficult. In S3 Fig, we see several randomly chosen input alignments, along with the true and predicted introgressed haplotypes. These results illustrate IntroUNET's ability to recover introgressed haplotypes with high accuracy in this simulated scenario.

Next, we examined the impact of sample size, the number of training examples, and the size of the input alignment (i.e. the number of polymorphisms), on IntroUNET's performance. In S4 Fig, we show the trajectories of our loss function, calculated on both the training and validation sets, over the course of training. We observe only a modest decrease in loss calculated on the validation set when the sample size per population is increased from 32 to 64, and further increasing to 128 yields no improvements. As we increase both the number of training examples ($1 \times 10^3$, $1 \times 10^4$, $1 \times 10^5$) and the length of the alignment (64, 128, and 256 polymorphisms), validation loss decreases continually albeit with diminishing returns. We observed that as we increase the size of each of these properties of the input/training set, the training time increases approximately linearly. GPU memory requirements also increase as the input dimensions grow. We therefore used an image size of 64 individuals per population and 128 polymorphisms, as this seems to represent an acceptable balance between accuracy and computational efficiency.

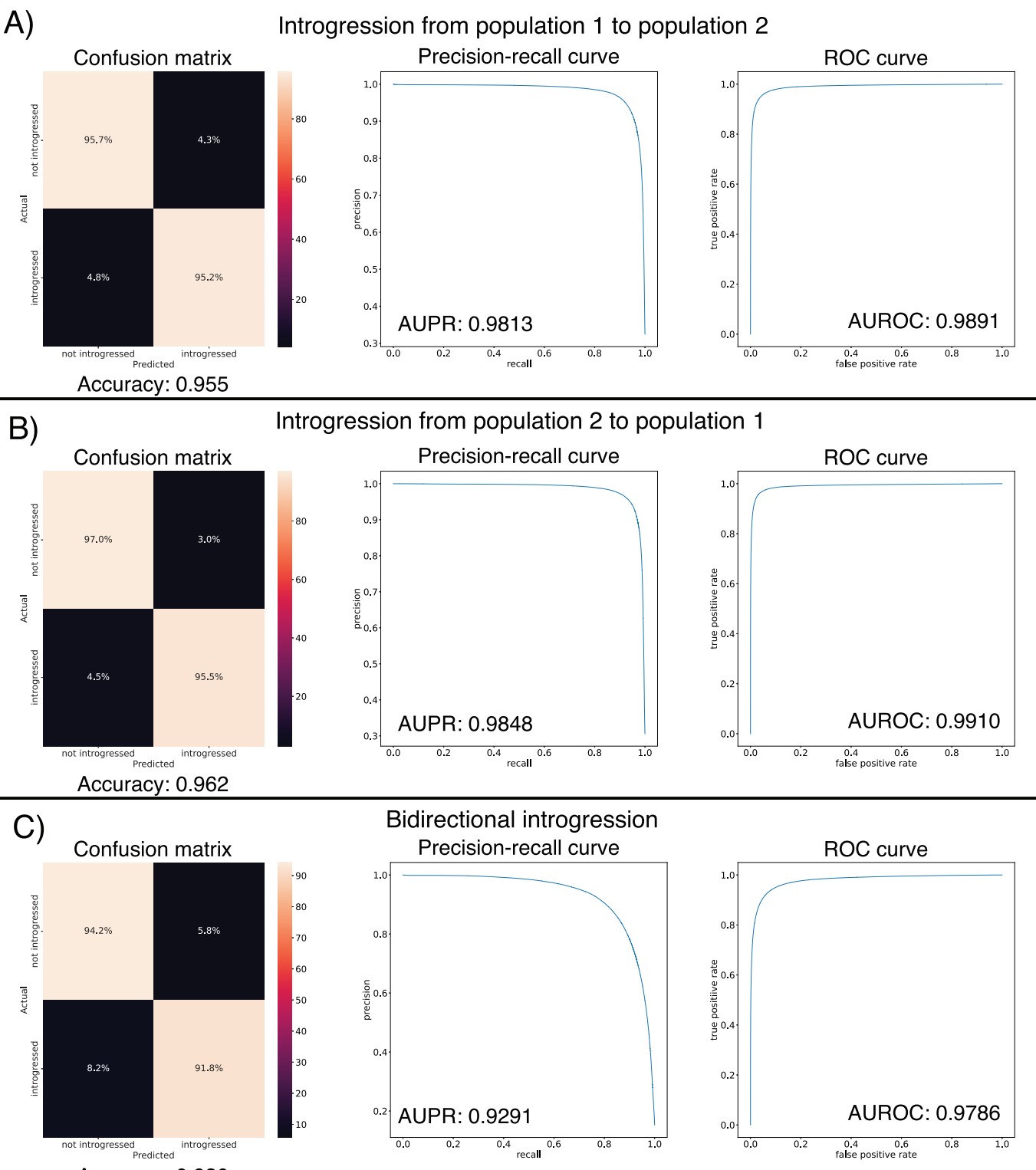

**Fig 4. Accuracy of `IntroUNET` on the simple introgression scenario.** (A) Confusion matrix, precision-recall curve, and ROC curve showing `IntroUNET`'s accuracy when trained to detect introgression in the direction of population 1 to population 2 and tested on data with introgression in this same direction. (B) Same as (A), but for a network trained and tested in data with introgression from population 2 to population 1. (C) Same as (A) and (B), but for bidirectional introgression. Note that all of these metrics evaluate `IntroUNET`'s ability to accurately identify individual alleles (i.e. a prediction is made for each pixel in each input image in the test set, and the accuracy of this prediction is evaluated).

## 3.2 IntroUNET performs well under scenarios of model misspecification

Like all model-based methods for population genetic inference, with IntroUNET there are concerns that the user-specified model may differ enough from the true evolutionary history of the data so that performance may be significantly impacted. We therefore examined the impact of several types of model misspecification on IntroUNET's accuracy: misspecification of the population split time and the introgression times/direction, the presence of direct selection for or against introgressed haplotypes, and the presence of unmodeled background selection.

Given that there may be uncertainty surrounding the population split time and also the time at which secondary contact resulting in introgression may have begun, we asked how IntroUNET's performance was affected if these parameters were incorrectly specified during training. We generated a grid of simulated datasets for all combinations of 5 values each of the migration time upper bound and split time parameters. Specifically, the split time was set to $2N$, $3N$, $4N$, $5N$, or $6N$ generations ago, and the migration time upper bound, expressed as a fraction of the split time, was set to 0.1, 0.2, 0.3, 0.4, or 0.5. This yielded a total of 25 parameter combinations, and a training and test set was generated for each. A version of IntroUNET was then trained on each of these training sets, and each trained network was then tested on all 25 test sets. As shown in S5 Fig, accuracy was acceptable in most cases of misspecification, and misspecification of the split time had only a modest impact on performance. Interestingly, accuracy suffered the most when the network was trained on a wide range of introgression times but only recent introgression had occurred. On the other hand, when trained only on more recent migration times, accuracy on data with a wider range of migration times remained high in most cases. This may imply that training on the lower end of one's confidence window of the migration times could be a reasonable strategy, but repeating this analysis on a wider range of demographic models may be required to confirm the effectiveness of this approach. Another form of misspecification with respect to introgression parameters is the direction if gene flow. For example, there may be cases where introgression is unidirectional but the user has applied a bidirectional version of IntroUNET. In S6 Fig, we show that the impact of this form of misspecification on accuracy is minimal (accuracy decreases from $\sim 96\%$ when the correct unidirectional network is used to $\sim 95\%$ when the bidirectional network is used).

Next, we asked whether direct selection on introgressed segments had a substantial impact on IntroUNET's accuracy. We did this by generating a new test set in the same manner as described for the bidirectional scenario evaluated in Fig 4, but this time giving selection coefficients to introgressed nucleotides. In S7 Fig, we show the impact of direct positive and negative selection on IntroUNET's accuracy. When introgressed segments are deleterious, as might be expected, we do not observe any decrease in accuracy. When introgressed segments are positively selected, accuracy is very similar to the neutral case when the selection coefficient per introgressed nucleotide is $1 \times 10^{-6}$ (92.9% accuracy versus 93.0% accuracy in Fig 4C), or approximately 0.00995 for a 10 kb introgressed segment (using multiplicative fitness effects as done by default in SLiM). When the selection coefficient is increased to $1 \times 10^{-5}$ (or $\sim 0.095$ for a 10 kb segment), accuracy decreases significantly, and a manual examination of simulated examples revealed that this was because introgression became so pervasive in this scenario that in many regions the majority of a population's sample was introgressed. In such cases the network would often invert the classifications, distinguishing between introgressed and non-introgressed pixels but swapping their labels (S8 Fig), because there is not basis to distinguish between the two populations in this scenario. Thus, we expect that IntroUNET will perform

reasonably well when introgressed alleles are subject to selection, but the introgression is not so rampant that populations cannot be distinguished from one another.

We also investigated the impact of background selection (BGS), the impact of linked negative selection on neutral diversity [96], on IntroUNET's accuracy. To do this, we used stdpopsim [81, 82] incorporate background selection (modeled after randomly chosen regions of the *D. melanogaster* genome) into our simulations of the same introgression scenario described above (see Methods for details). We generated a test set of 1000 regions each 1 Mb in length and ran IntroUNET on the central 100 kb of each region. As shown in S9 Fig, the presence of unmodeled BGS did increase IntroUNET's false negative rate substantially, but the false positive rate decreased slightly S9 Fig. We note that in practice, demographic inferences will be biased by BGS in such a way that will cause them to reproduce some of the impacts of BGS on diversity (e.g. an underestimation of $N_e$ to recapitulate the reduction in diversity caused by BGS; see Figure 1 from [97]). This is especially so in species with genomes dense with functional elements like *Drosophila*, where roughly half of intergenic DNA is subject to direct purifying selection and thus the impact of BGS is most likely pervasive. Thus, the scenario examined here where BGS had no impact on the training data but was present in the test data is probably more pessimistic than would be the case in practice. Furthermore, there does not appear to be a strong impact of recombination rate on accuracy S10 Fig. This was the case for both the average recombination rate in central 100 kb of the simulated region, or across the simulated region as a whole. Thus, it appears that neither BGS nor recombination rate variation will cause IntroUNET to produce an excess of false positives, although the former may cause IntroUNET to underestimate the extent of introgression.

### 3.3 `IntroUNET` can handle unphased data with minimal loss of accuracy

By default, IntroUNET requires phased haplotypes as input. This may be an onerous requirement for systems in which phased haplotypes cannot be easily obtained or inferred. However, we have previously shown that deep learning algorithms can make accurate inferences from population genetic alignments without phased data for other problems [47, 98]. Most notably, Flagel et al. were able to infer recombination rates not only without phased haplotypes, but even without genotypes for simulated autotetraploid populations, with almost no loss in accuracy [47]. We therefore trained a modified version of IntroUNET that takes diploid genotypes as input, using the same training data as for the bidirectional introgression scenario examined above but with pairs of haploid genomes from the same population combined to form diploid individuals, and with genotypes represented by 0 (homozygous for the ancestral allele), 1 (heterozygous), or 2 (homozygous derived). In S11 Fig, we show that this unphased version of IntroUNET experiences only a very minimal drop in accuracy compared to the phased version applied to this same task: the area under the ROC curve drops from 0.979 to 0.974, the area under the precision-recall increases slightly from 0.929 to 0.945, and balanced accuracy drops from 0.930 to 0.920.

### 3.4 Reference-free inference of archaic local ancestry

Having demonstrated the efficacy of the IntroUNET on a simple scenario of introgression between two sampled populations, we next sought to investigate its performance and versatility by addressing a more challenging problem: detecting introgression from an unsampled, or "ghost", population. A recent paper from Durvasula et al. presented a novel method for identifying regions of a genome that are introgressed from an archaic ghost population using two population genomic samples: a sample from a population that received genetic material via introgression from the ghost population, and a reference sample from a population not

thought to have experienced significant introgression from the ghost population [40]. Following Durvasula et al., we refer to this as a reference-free scenario because there is no reference panel from the donor population (although there is a non-introgressed "reference" population). Such an approach can be used to identify alleles in the human genome that trace their ancestry to archaic human species such as Neanderthals or Denisovans. Again, we trained and tested our method using simulations, this time generated under Durvasula et al.'s model which was motivated by the task of identifying Archaic introgression from Neanderthals to modern humans [40] as described in the Methods. The two main differences between this network and that described in the previous section is that the two channels of our network's input correspond to the recipient and reference populations, respectively, and the output has only a single channel denoting whether a given allele was introgressed from the ghost population to the recipient or not (see Fig 3B for example input and output). For this problem, we also compared IntroUNET's performance to that of ArchIE, the logistic model created by Durvasula which uses a vector of statistics to infer whether a given individual contains introgressed alleles from the ghost population. Note that by averaging predictions across sliding windows, ArchIE can be used to obtain segmentations similar to those produced by IntroUNET (see Methods and [40]).

We find that IntroUNET and ArchIE perform similarly on this problem. The metrics reported in Fig 5 suggest that IntroUNET has slightly better accuracy than ArchIE on this task, and this is supported by an examination of the ROC and precision-recall curves also shown in Fig 5. However, we note that the confusion matrices shown in Fig 5 reveal a higher false positive rate for IntroUNET than ArchIE, with a higher false-negative rate for ArchIE. Given that in cases of rare introgression, false positive rates will be of greater concern, this result suggests that a more stringent classification threshold may be necessary in this scenario—an option that users can easily adjust from the IntroUNET command line. Finally, we note that IntroUNET achieved accuracy metrics substantially lower than for the scenario tested in the previous section, where data from both the recipient and donor populations are available, underscoring that this archaic ghost introgression scenario is a more difficult task. Nonetheless, IntroUNET's relative effectiveness on this problem demonstrates that IntroUNET is a versatile framework for detecting introgression, as it can readily be adapted to very different scenarios without the need to adopt a different set of specialized statistics for the task at hand. Example segmentations produced by IntroUNET and ArchIE are shown in S13 Fig.

## 3.5 IntroUNET accurately detects introgressed haplotypes between *D. simulans* and *D. sechellia*

We had previously developed a machine learning method, called FILET, for detecting introgressed loci and applied it do data from *D. simulans* and *D. sechellia*, training it on a demographic model that we estimated from these two species [39]. While this effort revealed genomic regions that were introgressed, predominantly in the direction of *D. simulans* to *D. sechellia*, FILET can only predict whether a given window is introgressed, and cannot reveal the boundaries of introgressed haplotypes and the individuals having them. Thus, we sought to revisit this dataset as both a real-world proof-of-concept for our new method, and also to characterize patterns of introgression between these two species in greater detail.

Because the joint demographic history of these two species of *Drosophila* is considerably more complex than those of the test cases considered above, we first sought to evaluate IntroUNET's performance on data simulated under a demographic model estimated from these data. As we had previously [39], we modeled the demographic history of *D. simulans* and

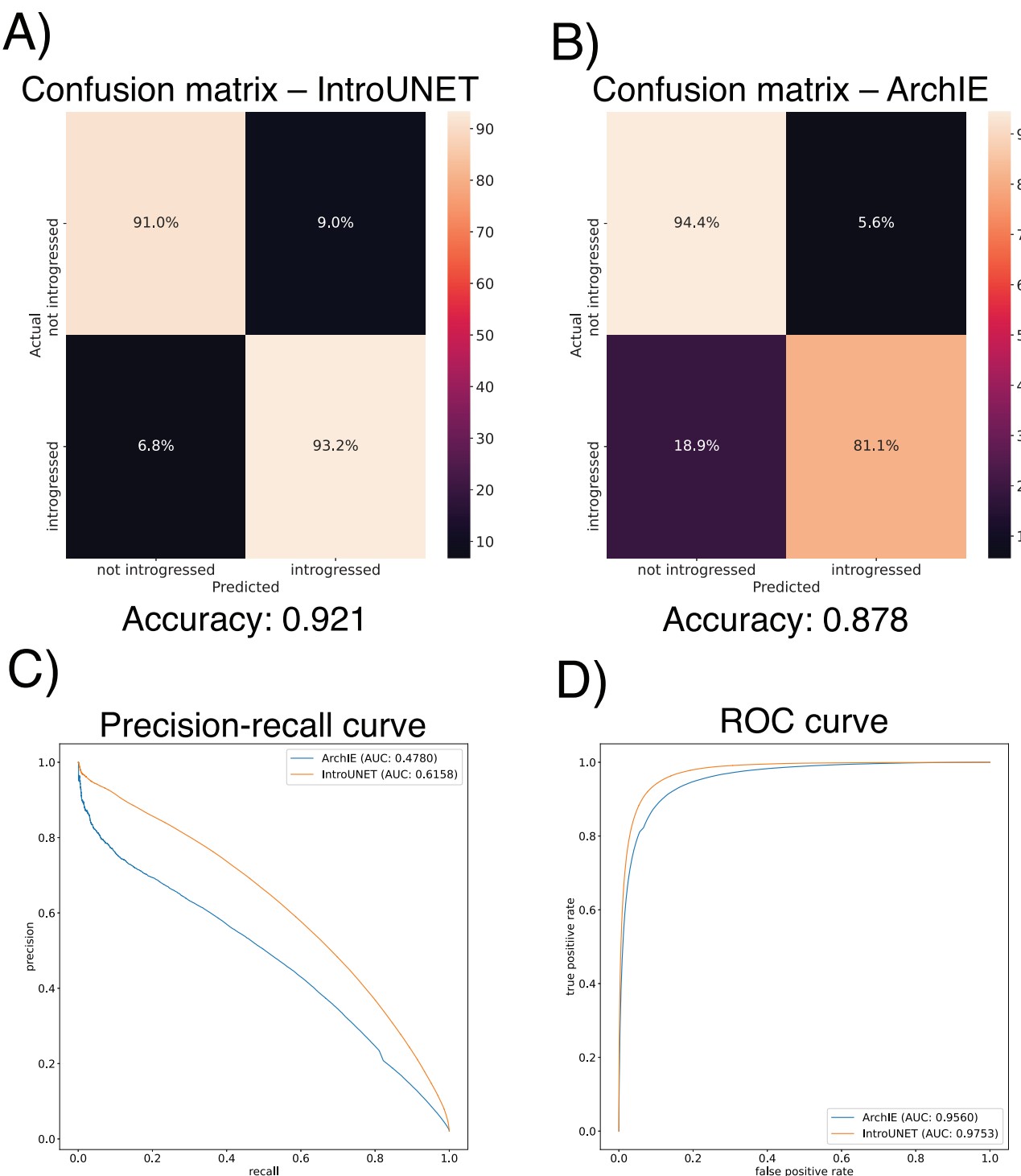

**Fig 5. Accuracy of `IntroUNET` and `ArchIE` on the archaic ghost introgression scenario.** (A-B) Confusion matrices, (C) precision-recall curves, (D) and ROC curves showing `IntroUNET`'s and `ArchIE`'s [40] accuracy when trained to detect introgression from a ghost population to a recipient population when given population genetic data from the recipient population and a closely related reference population.

*D. sechellia* using a two-population isolation-with-migration model, allowing for exponential population size change in the two daughter populations, but having a constant ancestral population. To account for uncertainty in the estimated parameters of this model, we ran 100 bootstrap replicates using ∂a∂i as described in the Methods, filtered replicates with low likelihoods, and simulated an equal number of examples under each of the remaining 43 inferred demographic models. As before, we omitted continuous migration from these simulations, only including single pulse migration events in simulated examples with introgression in order to control the timing of migration and track introgressed alleles (following [39]). Because we previously found that gene flow between these two species was primarily in the direction of *D. simulans* to *D. sechellia*, we modified our IntroUNET to detect introgression in this direction only (i.e. each allele in each chromosome was classified either as not introgressed, or introgressed in the direction of *D. simulans* to *D. sechellia*). We then used these simulations to train and evaluate IntroUNET as described in the Methods (see Fig 3C for example input and output).

We found that the IntroUNET was able to identify introgressed alleles in this simulated scenario, although the accuracy was not quite as high as observed for our simple test case described above (∼90% in Fig 6 versus ∼95% for the unidirectional cases in the simple scenario in Fig 4). We did observe that segmentations tended to be quite accurate within simulated regions that had experienced introgression (see S14 Fig for examples), but a substantial number of false positives were produced in simulated regions that had no introgression. We therefore limited our analysis to regions of the genome that we previously showed to be affected by introgression (see Methods and [39]). We also note that accuracy improved from 89.6% to 91.4% after recalibrating IntroUNET's probability estimates using Platt scaling [95]; an examination of the calibration curve (S15A Fig) and the confusion matrices before and after recalibration (Fig 6A and 6B) reveal that the uncalibrated version of IntroUNET was overestimating the probability of introgression, and that recalibration corrected this (S15B Fig). We therefore used the recalibrated version of IntroUNET in our analysis below.

We reexamined the 246 10 kb windows that we previously found to be introgressed from *D. simulans* to *D. sechellia*, using IntroUNET to identify introgressed haplotypes in these regions. These windows contained an average of 2086.5 SNPs, of which 705.3 (33.4%) on average were inferred to be in at least one introgressed block (Methods); we refer to these as introSNPs. At these introSNPs, an average of 3.3 *D. sechellia* samples were inferred to have an introgressed haplotype (Methods). We next asked whether the frequencies of introgressed haplotypes differed between genic and intergenic regions of the genome. We found that introgressed haplotypes were typically found at lower frequency in genic than intergenic regions (3.2 vs. 3.7 genomes inferred to be introgressed at the average introSNP in genic and intergenic regions, respectively), consistent with the action of purifying selection against introgressed alleles. The estimated distributions of the frequency of introgressed haplotypes in genic and intergenic regions are shown in Fig 7a.

Although the above results are consistent with the notion that introgression is often deleterious, a region on the right arm of chromosome 3 (chr3R:4539900–4769900) was previously shown to have an especially large block of introgressed alleles [39] with at least one of these introgressed alleles experiencing strong positive selection within the *D. sechellia* population [99]. If this were the case, we would expect introgressed alleles in this region to be at especially high frequency, as neutral or even slightly deleterious introgressed alleles would have hitchhiked to higher frequency along with the sweeping allele(s). We find that this is indeed the case, with the average introgressed haplotype found in 3.7 individuals within this region, versus an average of 3.2 outside of this region (Fig 7b). More strikingly, we observe a marked increase in the fraction of intermediate-frequency introgressed alleles in the region

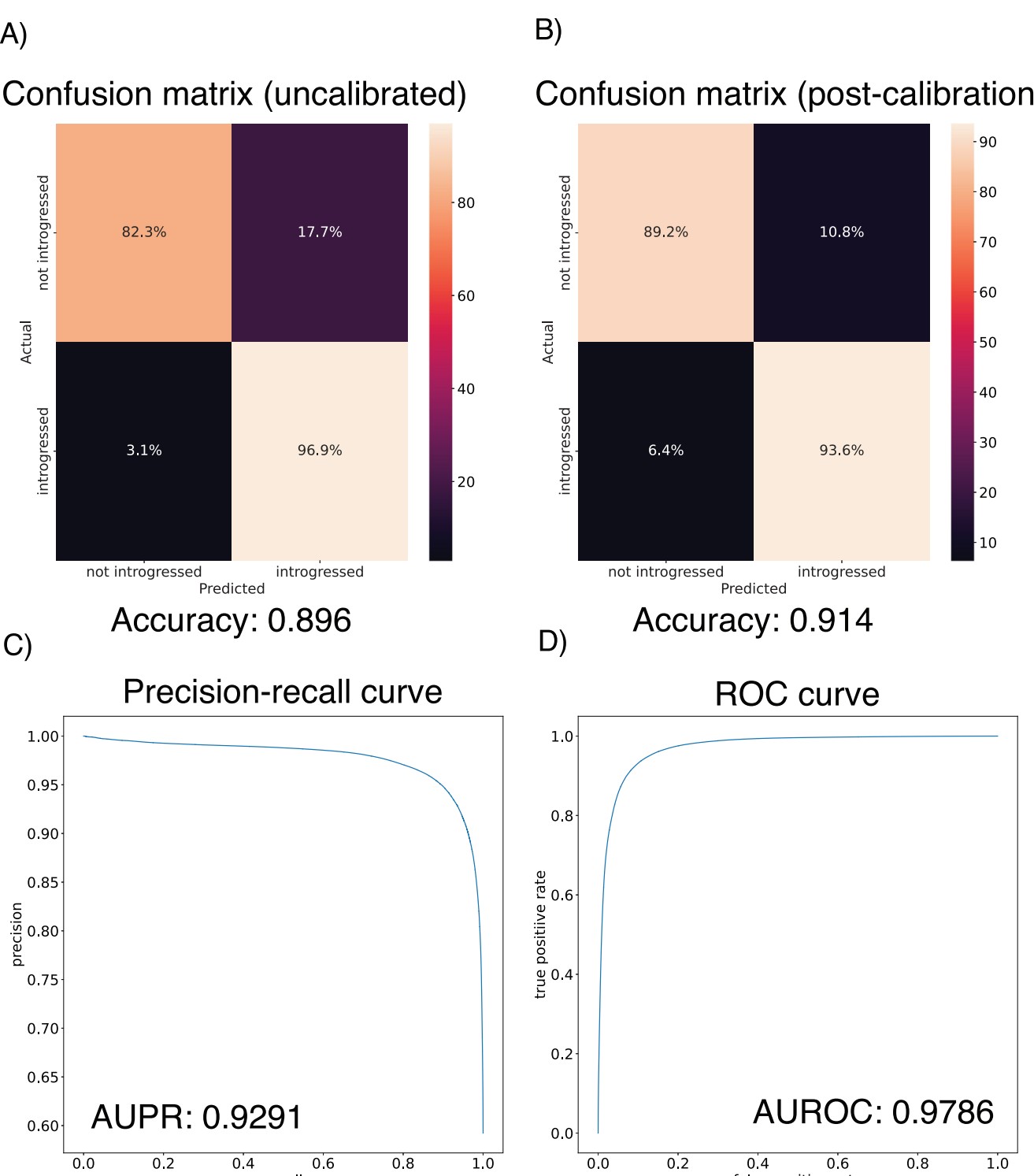

**Fig 6. Accuracy of IntroUNET on the *Drosophila* introgression scenario.** (A) Confusion matrix for the uncalibrated IntroUNET when applied to test data simulated under the *Drosophila* scenario as specified in the Methods. (B) Confusion matrix for the reclibrated IntroUNET. (C) and (D) show the Precision-recall and ROC curves for the *Drosophila* IntroUNET; note that these curves are not affected by recalibration.

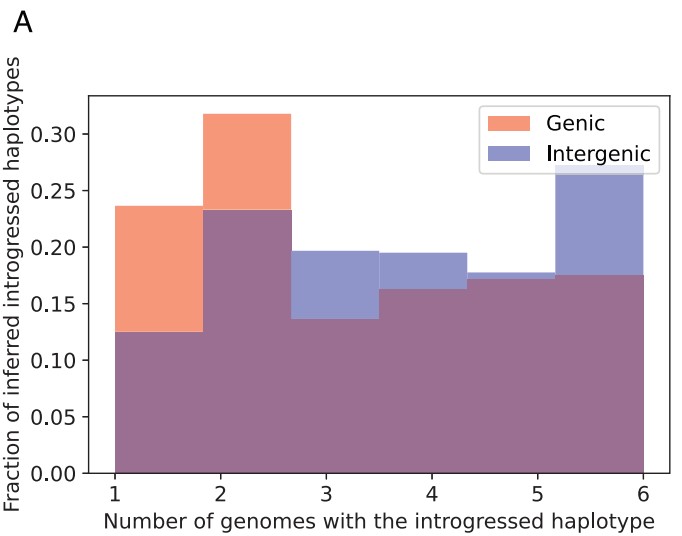 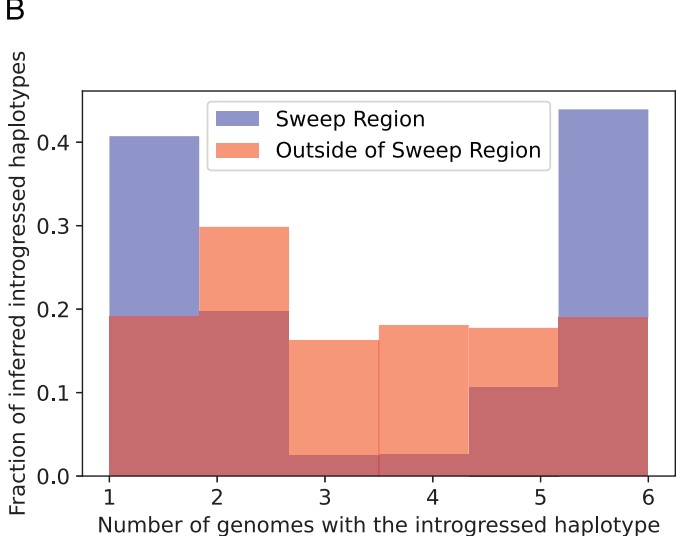

**Fig 7. The distributions of predicted frequencies of introgressed haplotypes in A) genic (red) and intergenic (blue) regions across the genome and B) the sweep region on chr3R (blue) and other regions of the genome (red).**

surrounding the sweep, a finding that is broadly consistent with theoretical predictions of allele frequencies located at moderate recombination distances away from the sweep (see Figure A3 from [100]). In S16 Fig, we show IntroUNET's predictions for three example windows near the sweep, but far enough away to contain appreciable amounts of diversity, as well as three randomly selected introgressed windows that appear to lie outside of the region affected by adaptive introgression. Finally, we observed a dramatic decrease in IntroUNET's predicted lengths of introgressed haplotypes at increasing distances away from the center of the sweep region (an ~12-fold reduction in mean haplotype length between the regions nearest to and furthest from the sweep shown in S17 Fig), as would be expected around the site of an adaptive introgression event [101]. The fact that IntroUNET infers the presence of long, high-frequency introgressed blocks within a region affected by adaptive introgression, but shorter and lower frequency of introgressed alleles in coding regions elsewhere in the genome, is consistent with biological expectations and implies that IntroUNET is able to make accurate inferences on real as well as simulated data.

### 3.6 Sorting via seriation improves IntroUNET's accuracy

As described in the Methods, IntroUNET preprocesses its input by first ordering the individuals of one population and then matching individuals in the second population to those in the first, and the choice of algorithms used for these steps (or whether to perform them at all) will therefore influence IntroUNET's output. To gain some insight into what effect the decision to sort rows of the input alignment and the choice of sorting metric (i.e. the measure of distance between two sequences) might have on the ability of an FNN to detect introgressed haplotypes, we trained our architecture on the simulated *Drosophila* and archaic ghost-introgression datasets both without sorting and after seriating with various choices of metric. For the set of metrics, we used many common distance metrics used to calculate distances between vectors: L1 or city-block distance, Pearson's correlation coefficient, the cosine distance, the Dice coefficient, or Euclidean distance, standardized Euclidean distance, and the Russel-Rao metric. In figure S18 Fig we show the training and validation loss obtained when

training repeatedly on the same dataset but using different distant metrics or without seriating at all. We find that sorting and the choice of distance metric have a sizeable effect on the ability of our neural network's final accuracy as measured by validation loss. In particular, the Dice coefficient metric performed the best, with cosine, city block, and Euclidean distances also performing well. Using standardized Euclidean distance or omitting the sorting step entirely both produced notably worse performance than sorting with the other metrics. These results underscore the importance of sorting data and choosing an appropriate distance metric (e.g. the Dice coefficient) when using IntroUNET.

We did not investigate whether the choice of which population is first sorted and which population is linear-sum-assigned to the former, or whether the choice of distance metric used in the linear-sum-assignment step has any effect on the efficacy of our architecture for the problems discussed here. For each problem, we chose the population that seemed to have the most diversity to be seriated and linear-sum-assigned the other to it.

### 3.7 Computational cost and implementation

Here we examine the computational cost of training and applying IntroUNET. While the simulation of synthetic alignments comes at low cost, both the training algorithm and the application of seriation to the individuals of one or more populations are more costly. We address these in turn below.

First, we compared the training times for IntroUNET and ArchIE. We trained ArchIE via the R code provided by [40] on a AMD EPYC 7413 CPU clocked at 3 GHz and IntroUNET method via an NVIDIA RTX 3090 GPU. ArchIE took roughly 7 minutes to train but large amounts of CPU RAM (>32 Gb) as the method requires the entire dataset be made available to R. Our method took $\sim$9.65 hours to train on this problem until convergence.

Next, we examined the computational speed of seriation, and compared this to the feature vector calculations required by ArchIE. Although the seriation problem is NP-hard, computationally tractable algorithms for approximately solving this problem do exist, and it is this approach that we take here (Methods). We found that for both the *Drosophila* scenario ($n = 32$) and the archaic introgression scenario given by [40] ($n = 112$) it took an average of 1.002 seconds to seriate one of the populations in a 128-segregating site alignment. This was calculated over 100 simulation replicates on an Intel Core i9–9900K CPU @ 3.60GHz. For comparison, it took an average of 0.08976 and 0.01701 seconds to simulate one replicate under the *Drosophila* and archaic introgression models [40] respectively, although other scenarios and simulators may require more computation. It is worth noting that the difference in sample sizes in the two examples we benchmarked (112 vs 32) did not increase average computation time (within one millisecond). We also note that the seriation routine that we used greatly outpaced the average computation time for calculating the features used by ArchIE for detecting introgression [40]—the latter took an average of 5.715 seconds to compute for each window simulated, although we took these functions directly from the repository provided and did not attempt to optimize them. Thus, although IntroUNET formats its input much faster than ArchIE, we note that for both of these methods this step can be accelerated dramatically by formatting subsets of the training set in parallel on a high-performance computing cluster, and that the routines we package with the repository to do so use MPI to accomplish this.

In order to seriate an alignment, one must define a distance metric and compute the pairwise distance matrix for all pairs of sequences in the population sample to be sorted. If we have $n$ elements, each a $p$-dimensional vector then we have a complexity of $n(n-1)/2$ multiplied by the complexity of computing the distance metric itself, which for many common metrics is

simply a linear function of $p$. For instance, the computational complexity of calculating the Euclidean distance matrix is simply $\frac{3pn(n-1)}{2}$. As described in the Methods, we used the Kuhn-Munkres algorithm to perform linear matching between the two population samples in each input. Although this algorithm has a complexity of $\mathcal{O}(n^4)$, it takes on the order of a millisecond to terminate for the relatively small sample sizes considered in this paper when benchmarked on the same CPU mentioned above.

In short, although our method requires more training time than ArchIE, the computational cost is reasonable: when the simulation, formatting, and training steps are considered together, the entire training process can be completed on a compute cluster within one day, provided GPU resources are available. A tabular summary of the running times of the machine learning methods examined here is found in section 4.

## 4 Discussion

It has been noted that at least 10% of species experience hybridization, creating the potential for introgression [102]. Thus, in some species a sizeable fraction of the genome may be affected by cross-species introgression, and a number of methods have therefore been developed to detect introgressed loci (e.g. [14, 24, 39]). We have created a tool, called IntroUNET, that adapts a powerful deep learning method for semantic segmentation to the task of detecting alleles that have introgressed from one population to another by examining patterns of variation within an alignment consisting of samples from two populations undergoing recent gene flow with one another. We showed that IntroUNET can accurately recover introgressed haplotypes, and which individuals harbor them, in simulated data. With minimal adjustment, our method can be adapted to detect archaic "ghost" introgression by examining a two-population alignment consisting of the recipient population, and a "reference" population experiencing comparatively little to no gene flow. On this task, IntroUNET performs at least as well as ArchIE, a machine learning method that uses a set of features engineered for this specific task. This relatively straightforward modification to successfully attack a different introgression-detection task demonstrates the flexibility of IntroUNET, and deep learning approaches in general [47]. We additionally note that in some scenarios a non-introgressed "reference" population may not be not be available, and other methods that do not require this reference would be required in such cases [39, 87, 103]. Future work could examine whether the IntroUNET framework could be successfully applied to this scenario.

Importantly, we showed that IntroUNET is relatively robust to several forms of model misspecification: erroneous population split times and migration times (S5 Fig), violations of our model's assumptions of neutrality of introgressed haplotypes (S7 Fig), and the presence of unmodeled background selection and recombination rate variation (S9 and S10 Figs). The first of these analyses is also relevant to another type of misspecification that could potentially impact our analysis of the *Drosophila* dataset: for this analysis we estimated continuous migration parameters between *D. simulans* and *D. sechellia*, but we trained IntroUNET using a modified version ms that can only track introgressed alleles resulting from pulse migration events. Because introgression in these data appears to be relatively rare, affecting only a minority of the genome [39, 56], those loci that are affected typically will only have experienced a single introgression event—note that the example regions in S16 Fig are consistent with this. Because IntroUNET is trained on simulations that experience only a single introgression time within a given region, but the migration time varies across regions, this form of misspecification is a problem only inasmuch as it may result in migration times differing between the true and simulated data. Thankfully, our results in S5 Fig suggest that IntroUNET will typically perform quite well in the presence of such misspecification. In addition, we note that

IntroUNET can be used on unphased data and experiences only a small decrease in performance relative to the phased case (S11 Fig). This property, combined with its robustness to misspecification, make it a versatile tool that can be applied to model and non-model systems alike.

We were also able to apply IntroUNET to a *Drosophila* dataset that we examined previously [39]. This dataset consists of *D. simulans* samples from mainland Africa, and *D. sechellia* samples from the Seychelles, an island nation where *D. simulans* is also present and where hybridization between the two species is known to occur [104]. It had previously been shown that there was substantial introgression between these species [56], and we had found that this gene flow was predominantly in the direction of *D. simulans* to *D. sechellia* [39]. Detecting introgression is somewhat more challenging in this data set than in the simple two-population scenario that we examined initially, most likely because of the relatively recent split time between these two species [39], resulting in much-reduced levels of diversity in this species. Nonetheless, IntroUNET performed quite well on data simulated under this demographic scenario, and when applied to the real dataset it also revealed two key patterns that were consistent with expectations, underscoring IntroUNET's practical utility. First, we observed lower frequencies of introgressed material in genic versus intergenic regions, consistent with the notion that introgression is often deleterious [6]. Second, IntroUNET predicted much higher frequencies of introgressed alleles within a region of the 3R arm that was previously shown to be affected by adaptive introgression [99], as expected under a scenario where the hitchhiking effect will cause neutral introgressed alleles that are linked to the selected allele to hitchhike to higher frequencies [100, 105]. This suggests that IntroUNET correctly identifies introgressed haplotypes even if one of the core assumptions of our training process—selective neutrality—is violated.

Although IntroUNET performed well on our *Drosophila* dataset overall, our analysis did reveal two limitations that future advances may be able to address. First, given that detecting introgression in the direction of *D. simulans* to *D. sechellia* is especially challenging (see [39]), we found that it was necessary to limit our analysis to regions that showed strong evidence of introgression. This is because, on our simulated test data for this scenario, IntroUNET infers some individuals/alleles as introgressed even in windows where no introgression is present. However, in regions where introgressed alleles are present, IntroUNET was often able to detect them accurately (S14 Fig). To improve the usability of our method, we have therefore included in the IntroUNET package a neural network for classifying genomic windows as having experienced introgression or not (Methods), in the same vein as the approach taken by Flagel et al. (see Figure 4 from [47]). As shown in S19 Fig, this classifier, when trained to detect genomic windows that have experienced introgression between *D. simulans* and *D. sechellia*, is highly accurate. We recommend that users test the performance of IntroUNET on simulated data prior to analyzing real data, and if they observe unsatisfactory performance in non-introgressed regions in simulated test data, running our window-based classifier as a first step may allow users to proceed with accurate segmentation on regions that appear to have experienced introgression, as we were able to do for our *Drosophila* dataset.

Second, we also observed that, on data simulated under our *Drosophila* demographic model but experiencing no selection, IntroUNET occasionally produced blocks of sites where nearly every individual was predicted (incorrectly) to have introgressed alleles. This may be a consequence of the low level of diversity in the *D. sechellia* population—when all individuals are nearly identical we might expect false positive predictions to be propagated across the entire alignment. Given that we were primarily concerned with regions where only a subset of genomes had introgressed alleles, as manual examination of introgressed loci from Schrider et al. [39] had revealed that introgressed haplotypes typically appeared to be present at lower

frequencies, this issue was addressed by simply filtering out all sites where the introgressed allele was predicted to be present in at least half of our *D. sechellia* sample. Another observation we made during our analysis of the *Drosophila* introgression scenario is that IntroUNET's raw sigmoid outputs do not give well-calibrated estimates of the probability of introgression. We were able to resolve this via Platt recalibration, which produced far better calibrated probability estimates (S15 Fig). We therefore recommend this recalibration step for any analyses that hinge on the accuracy of the introgression probability estimates produced by IntroUNET, and we have incorporated this functionality into the IntroUNET software.

We demonstrated that a semantic segmentation framework can be successfully adapted to solve population genetic problems. Our method, IntroUNET, uses a variant of the U-net architecture [69], called Unet++ [68], to accurately identify introgressed alleles under a user-specified demographic history (specified during the simulation of training data) and sampling scheme. In addition to its impressive accuracy and flexibility, IntroUNET is computationally efficient, requiring on the order of a day or less for the entire training process when a high-performance computing cluster is available (to accelerate data simulation and sorting), with very rapid downstream prediction as well. Indeed, IntroUNET can feasibly be trained and run on a single consumer grade computer. However, we note that if experimentation is needed to identify the optimal neural network architecture and hyperparameters, then a cluster with multiple GPU compute nodes may be needed to make the task time-feasible. We also note that our method uses a heuristic to deal with the fact that the ordering of individuals in a population genomic alignment is generally not meaningful, and there are therefore many possible image representations of a single alignment: sorting via seriation. Although this approach is relatively fast and improves classification accuracy, it is not guaranteed to produce the optimal ordering of individuals for the detection of introgressed haplotypes. Future work may obviate the need for sorting altogether by using more efficient methods that may not require sorting, such as permutation invariant neural networks [45], or Graph Neural Networks (reviewed in [106]) based on inferred tree sequences [107, 108].

## Supporting information

**S1 Fig. A diagram of the chosen residual block structure shown with the dimensions for the first convolution for an initial size of two-populations by 64 individuals by 128 polymorphisms.**
(PDF)

**S2 Fig. Loss function value trajectories calculated on training and validation data for versions of IntroUNET with different values of the label smoothing strength parameter *alpha*.** All tests were calculated on simulated examples of the simple bidirectional scenario described in the Methods. Note that for *alpha* = 0.1 training loss is higher than validation loss. This is because label smoothing is only applied during training, and smoothing increases loss by adding noise to the target *y* values.
(PDF)

**S3 Fig. Five randomly chosen example segmentations on simulations from our simple bidirectional introgression scenario.** Each example shows the input alignments for the two populations (labeled "pop 1" and "pop 2" respectively), the true introgressed alleles for these two populations (labeled "pop 1 (y)" and "pop 2 (y)" respectively), the introgressed alleles inferred by IntroUNET ("pop 1 (pred)" and "pop 2 (pred)"), and IntroUNET's inferred introgression probabilities (labelled "prob", and scaled according to the color bar shown with the third example). Alignments and introgressed histories, true and predicted, are shown in the same

format as in Fig 1.
(PDF)

**S4 Fig. Loss function value trajectories calculated on training and validation data for versions of IntroUNET with increasing sample sizes (32, 64, and 128 individuals per subpopulation), training set sizes (1000, 10000, and 100000 alignments), and window sizes (64, 128, and 256 polymorphisms).** All tests were calculated on simulated examples of the simple bidirectional scenario described in the Methods.
(PDF)

**S5 Fig. IntroUNET's accuracy when the split and migration times may be misspecified.** IntroUNET was trained on 25 different combinations of the population split time and the upper bound of the range of possible introgression times (with the lower bound always set to zero). These simulations were performed in the same manner as described for the simple bidirectional model in the Methods, with the exception of these two parameters. Each heatmap in this grid shows the accuracy of one version of IntroUNET on each of the 25 test sets, and the parameter combination used to train that network is marked by a circle. For example, if the true split time is $2N$ generations ago and the true split time is 0.1 times the split time, one can observe the impact of misspecification on accuracy by comparing the top-left value in the top-left heatmap (i.e. no misspecification in this case) to the top-left value of all other heatmaps in the figure, which experience varying degrees of misspecification.
(PDF)

**S6 Fig. Performance of IntroUNET on the task of identifying introgression on a dataset where introgression is occurring in only one direction, but IntroUNET was trained to detect bidirectional introgression.** The test data here were the same as those examined in Fig 4A, but the network used to perform inference was the same as that used for Fig 4C.
(PDF)

**S7 Fig. The impact of direct selection on IntroUNET's performance.** The left column shows confusion matrices obtained when using the same trained network used for the simple bidirectional introgression scenario whose parameters are laid out in Table 1, and applied to data simulated under the same model but with introgressed nucleotides experiencing negative selection. The column on the right shows results when testing the same IntroUNET model on data where introgressed segments are positively selected. The values of $s$ represent the selection coefficient per introgressed nucleotide. Note that the shading represents the number of examples in each entry of the confusion matrix rather than the fraction of examples.
(PDF)

**S8 Fig. Five randomly chosen example segmentations on simulations from our simple bidirectional introgression scenario with relatively strong positive selection acting on each introgressed nucleotide ($s = 1 \times 10^{-5}$, or $\sim 0.095$ for a 10 kb introgressed segment).** Each example shows the input alignments for the two populations (labeled "pop 1" and "pop 2" respectively), the true introgressed alleles for these two populations (labeled "pop 1 (y)" and "pop 2 (y)" respectively), the introgressed alleles inferred by IntroUNET ("pop 1 (pred)" and "pop 2 (pred)"), and IntroUNET's inferred introgression probabilities (labelled "prob"). Alignments and introgressed histories, true and predicted, are shown in the same format as in Fig 1.
(PDF)

**S9 Fig. The impact of background selection (BGS) on IntroUNET's performance.** The confusion matrix shows IntroUNET's classification performance when applying the same

trained network used for the simple bidirectional introgression scenario whose parameters are laid out in Table 1 to data simulated under the the *D. melanogaster* BGS model specified in the Methods. Note that the shading represents the number of examples in each entry of the confusion matrix rather than the fraction of examples.
(PDF)

**S10 Fig. The impact of recombination rate on IntroUNET's accuracy under the *D. melanogaster* background selection model.** The left panel shows each simulation's accuracy, averaged across classified pixels from the central 100 kb of the simulated chromosome, as a function of the average recombination rate in that same window. The right panel shows each simulation's accuracy, averaged across classified pixels from the central 100 kb of the simulated chromosome, as a function of the recombination rate averaged across the entire simulated 1 Mb chromosome.
(PDF)

**S11 Fig. Performance of IntroUNET on the task of identifying bidirectional introgression using unphased genotype data.** The test data here were the same as those examined in Fig 4 (C), but here rather than predicted which haplotypes are introgressed, IntroUNET, when given a matrix of diploid genotypes, infers which diploid individuals have at least one introgressed allele at a given polymorphic site.
(PDF)

**S12 Fig. Diagram of the ghost introgression demographic model from [40], in which an unsampled archaic population splits off from the main population, before a pulse introgression event introduces alleles from this population into a sampled "Target" population, not an unsampled "Reference" population.**
(PDF)

**S13 Fig. Ten example segmentations on simulations from our archaic introgression scenario: 5 from IntroUNET (left) and 5 from ArchIE (right).** For each example we show the true and inferred introgressed alleles in the recipient population. For each method, both examples with and without introgression are shown.
(PDF)

**S14 Fig. Five example segmentations on simulations from our *Drosophila* introgression scenario.** Each example shows the input alignments for the two populations, the true and inferred introgressed alleles for the *D. sechellia* population, and IntroUNET's inferred introgression probabilities. Alignments and introgressed histories, true and predicted, are shown in the same format as in Fig 1.
(PDF)

**S15 Fig. Calibration curves showing the impact of Platt scaling on the accuracy of the introgression probability estimates produced by IntroUNET, calculated on the validation set from the *Drosophila* simulations (Methods).** A) The fraction of alleles falling within a given bin of IntroUNET's predicted probability of introgression that were in fact truly introgressed, prior to Platt recalibration. B) Same as (A), after recalibration.
(PDF)

**S16 Fig. Alignments and segmentations from six windows on chr3R in the vicinity of the locus of adaptive introgression (AI).** Three of the windows (left) are from outside of the region affected by AI, and the other three (right) are within the AI locus. Each example shows the input alignments for the two populations (labeled "**X** (*dsim*)" and "**X** (*dsech*)",

respectively), IntroUNET's inferred introgression probabilities ("$\hat{y}$ (probs)"), and the most probable class for each allele in each individual ("$\hat{y}$ (class)"). Inferred introgressed histories and introgression probabilities are shown in the same format as in Fig 1. Alignments are shown in the same format as for previous figures, with the exception that for *D. sechellia* which has a different color scheme in haplotypes that were inferred to be introgressed in order to highlight these regions of the alignment: blue for the ancestral allele, and red for the derived allele.

(PDF)

**S17 Fig. The lengths of introgressed haplotypes predicted by IntroUNET at increasing distances away from the adaptive introgression even on chr3R.** Introgressed haplotypes were defined as runs of consecutive SNPs classified as introgressed for a given individual. The sweep center was set to position 4624900, the center of the window with the lowest level of diversity in this region (data from [39]).

(PDF)

**S18 Fig. Results of training the same architecture on data seriated via different distance metrics, as well as using unsorted data (i.e. individuals within each population are arranged in the arbitrary order produced by the simulator), for the ghost-introgression problem (top row, panels A and B) and the *Drosophila* demographic model (bottom row, panels C and D).** These plots show the values of training (A and C) and validation (B and D) loss over the course of training. Validation loss is usually lower than training in the case of *Drosophila* because label smoothing was applied to the training data for the purposes of regularization, but not to the validation data.

(PDF)

**S19 Fig. Confusion matrix showing performance of a classifier that detects genomic windows that have experienced introgression, trained and evaluated on data simulated under the *D. simulans*-*D. sechellia* scenario as described in the Methods.**

(PDF)

**S1 Table. Bootstrap parameter estimates for the *D. simulans* and *D. sechellia* joint demographic model obtained via ∂a∂i.** The parameters of the model are the ancestral population size ($N_{ref}$), the final population sizes of *D. sechellia* and *D. simulans* ($N_{0-sech}$ and $N_{0-sim}$), the initial population sizes ($N_{sech}$ and $N_{sim}$), the population split time ($t_s$), and the backwards migration rates ($m_{sim \rightarrow sech}$ and $m_{sech \rightarrow sim}$). Note that parameter estimates are shown for each bootstrap replicate for which our optimization procedure succeeded (Methods), but only those with log-likelihood scores greater than −1750 were used to simulate training data.

(PDF)

**S2 Table. CPU/GPU time estimates for accomplishing the experiments in the paper.** The simulation and formatting results for this table were computed from a small sample of 430 replicates over 4 cores of an Intel Core i9–9900K CPU @ 3.60GHz and then scaled to give estimates for $10^5$ replicates in each case. The Formatting column lists the time estimates for alignment sorting (for IntroUNET) and statistic calculation (for ArchIE). The Discriminator and Segmenter columns list the training times for classifying entire windows and for identifying introgressed haplotypes, respectively. In the GPU columns the estimate is simply the run time for the training described. The training of the neural networks was done on an NVIDIA A40 GPU, and we found that VRAM usage was <12Gb in all cases. We note that the ArchIE method computes statistics over the entire simulated window (224.64 on average in our simulated 50kb windows) whereas our method only formats a small sequential sample of

SNPs from each replicate (192 for the Archaic introgression program). Below the time estimates for simulation are the simulated region size, and below the formatting times are the resulting "image" size or (populations, individuals, sites) and for the case of `ArchIE`, the window size in base pairs. Note that we do not include the time for execution on data after training, but we observed that classification times for all are generally negligible (although the sorting/statistic calculation steps must be performed first and these can be costly as shown in the Formatting section).
(PDF)

## Acknowledgments

We thank members of the Kern-Ralph Co-lab for feedback on preliminary results for this project.

## Author Contributions

**Conceptualization:** Dylan D. Ray, Lex Flagel, Daniel R. Schrider.

**Investigation:** Dylan D. Ray, Daniel R. Schrider.

**Methodology:** Dylan D. Ray, Lex Flagel, Daniel R. Schrider.

**Software:** Dylan D. Ray, Daniel R. Schrider.

**Validation:** Dylan D. Ray, Daniel R. Schrider.

**Visualization:** Dylan D. Ray, Daniel R. Schrider.

**Writing – original draft:** Dylan D. Ray, Daniel R. Schrider.

**Writing – review & editing:** Dylan D. Ray, Lex Flagel, Daniel R. Schrider.

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
