## [Decision Letter · Decision Letter 0]

27 Apr 2023

Dear Dr Schrider,

Thank you very much for submitting your Research Article entitled 'IntroUNET: identifying introgressed alleles via semantic segmentation' to PLOS Genetics.

The manuscript was fully evaluated at the editorial level and by independent peer reviewers. The reviewers appreciated the attention to an important problem, but raised some substantial concerns about the current manuscript. Based on the reviews, we will not be able to accept this version of the manuscript, but we would be willing to review a much-revised version. We cannot, of course, promise publication at that time.

If you decide to revise the manuscript for further consideration at PLOS Genetics, please aim to resubmit within the next 60 days, unless it will take extra time to address the concerns of the reviewers, in which case we would appreciate an expected resubmission date by email to plosgenetics@plos.org.

We are sorry that we cannot be more positive about your manuscript at this stage. Please do not hesitate to contact us if you have any concerns or questions.

Yours sincerely,

Nicolas Bierne

Academic Editor

PLOS Genetics

Kirsten Bomblies

Section Editor

PLOS Genetics

Dear Dr Schrider,

We have received three in-depth reviews of your manuscript entitled "IntroUNET: identifying introgressed alleles via semantic segmentation" (PGENETICS-D-23-00137).

The reviewers found your method for the identification of introgressed alleles via semantic segmentation and its application to be of interest. In my opinion, your paper has the potential to be an important contribution to the journal. However, the three reviewers have major concerns that must be addressed in a revision. Since the revision will have to include new analyses, this will require considerable work. In particular, you'll need to show that the method works for nonmodel species with less knowledge of their demographic history, and to explore more complex histories of divergence and secondary gene flow, including parameter combinations that makes the detection of introgression more difficult. You'll also need to clarify how the data are pre-processed.

I'll be looking forward to your revision of the paper. The same reviewers will review it, I expect.

Best regards,

Nicolas Bierne

Reviewer's Responses to Questions

**Comments to the Authors:**

Reviewer #1: In this paper, the authors employ current deep-learning approaches for semantic segmentation to develop a method IntroUNET for identifying tracts of introgressed genetic variation carried within individuals under (1) a model of population divergence with a single-pulse admixture event and (2) a model of archaic introgression from an unknown "ghost" population, demonstrating improved power relative to a previous ML method ArchIE designed explicitly for that task. They also apply the method to identify introgressed haplotypes in D. sechellia, here, accounting for previously-estimated population growth in their training, validation, and testing data. They recover previously-known haplotype patterns of introgression in this species demonstrating the accuracy of their method in real application.

The paper focuses on the technical aspect of the method and is very well written. I really appreciated the explainations in the methods section. I am familiar with ML methods, but by no means an expert, and with the current text and a bit of googling, I feel like I was able to understand how the method works, what it does, and how to interperet the results.

My comments below are about clarifying the text and assessing the performance of the method in the context of model mis-specification and realistic genomic architecture:

l 125 - 128: This sentence is a little dense. "Segmentation" needs to be defined more clearly / made more intuitive for the sake of this section. Maybe just say "In our case, we want to assign each pixel (SNP) as either "introgressed" or "not introgressed"."

l 194 - 195: It isn't clear from the paper to what extent this architecture in Figure 2 is decided by the authors. Is UNet++ a single pre-defined architecture? Or is this one of different flavours of Unet++? Is there any transfer learning being done here?

l 208: Maybe state directly that 'positive' means 'introgressed' and negative means 'not introgressed'. Is the choice of weighting $w_p$ a standard way of dealing with unbalanced ratios in this approach? Otherwise, how was this decided? Also, is yHat_n the prediction probability of the positive example rather than the label?

Table 1: Easier to just say that $t_s = 4N$ generations? In the 'simple' scenario, for bi-directional migration, is the migration rate drawn independently for the two populations? It surprised me to see so much gene flow in one but not the other direction. Also, is the probability of migration the forward or backward probability?

General: How much does mis-specification of the underlying demographic model affect the performance? In particular, I'd be interested to see how the method (as trained for the 'simple' scenario in the paper) performs when the test data set has a more recent split time.

l 294 - 297: Was the (not-downsampled) training data set the same for ArchIE as for IntroUNET?

l 310: It might be more accurate to say "Following [39], we used dadi to estimate..."

Between lines 323 and 324:

i) Were all 100 bootstraps used for sims, or only those listed in supp table 1?

ii) With regard to estimating continuous migration but using a pulse-admixture for training, validation, and testing... This is a bit funny to me. Those parameters estimated by dadi are all correlated with each other.

By taking those values out of the continuous-migration context and using them in a pulse-migration context, how well do they still represent the population under study? I think this is a good point for a model-misspecification test. How well does the network trained on pulse-admixture perform when tested on data simulated under the IM model?

Also, just out of curiosity, would it be possible to use the dadi-inferred IM demography directly for training purposes?

iii) label smoothing: This may be a standard thing for deep learning approaches, but it was confusing for me as a PGen reader. You tell me what it does, but I don't have any intuition as to why it helps.

l 326: It wasn't clear how this is helpful for training or what problems this helped to resolve. Is there a reason this and the Platt scaling are only used in the Drosophila analysis?

l 339-340: Do you have any intuition about why this happens?

l 391 - 392: What classification threshold was used in the manuscript? 0.5?

l 451 - 453: I may have misinterpereted this, but the frequencies of the introgressed haplotypes in the case of positive selection doesn't seem very high relative to the background. Aos, the fact that there's a U-shape to the distribution seems more important than the mean. Do you have any further insight or intuition about this?

l 479: As a reader, I want to know how feasible it is for me to run either of these methods given the resources I have access to. A supp table comparing the computation costs for archie and introUnet at the various stages would make this more clear.It would alos help to indicate the stages that can be parallelized in the respective approaches.

l 514 to 515: It isn't clear to me how much the sample size and number of snps (input image size) affects the cost to train the model.

l 546: With these image-processing ML methods, I tend to wonder how much the underlying genome architecture affects performance, and I think this could be investigated directly in this paper. I would like to see a small set of test simulations run in SLiM using the same population parameters here but including the D melangoaster genome annotation, genetic map, and distributions of deleterious fitness effects (they are in the standard popsim catalogue). Testing this data with the naive model would also help demonstrate the robustness of the method in real applications.

l 549: What types of advances would help resolve these issues? More extensive training with a broader range of parameters? Improvements to machine learning techniques themselves?

l 574-576: What experimentation did the authors do to decide on the architecture in figure 2? How sensitive is the method to the choice of architecture? If this matters, then how feasible is it for an IntroUNET user to find the optimal architecture and hyperparameters?

Reviewer #2: Ray and collaborators present a deep learning algorithm for semantic segmentation adapted to identify introgressed alleles in genome alignments. This work aligns with the mounting amount of studies on applying deep learning in genomics, highlighting its potential to provide evolutionary inferences in the genomic era.

From our point of view, this work’s most significant and original contribution is using semantic segmentation (based on an adapted version of UNET++) to tackle the problem of identifying introgressed alleles/haplotypes in an alignment of sequences. Compared to local ancestry inferences, significant improvements of IntroUNET are that a reference panel of donor populations is not required, and the introgression process is explicitly modelled in the simulations (although parameter values of the demographic scenario are not inferred). Moreover, it extracts genealogical information directly from the raw alignment of sequences by passing the summary statistics step, which a loss of information can accompany.

To say a word about the form, the writing style is generally clear with a very nice Introduction, the figures are well done, and the manuscript is pleasant to read. However, it is too technical in some places at the expense of biology, and some assumptions are not clearly stated, making the application of the method to actual data not so easy. Still, we are convinced that such a tool could bring numerous benefits to the field of population genomics. Below are a series of comments that we hope will improve the manuscript.

1. Scenarios

● In general, we are a little afraid that the scenarios modelled by the authors to test their IntroUNET method are a bit too simple to evaluate its ability to correctly identify introgressed haplotypes with actual empirical data. Indeed, in the first scenario, which is on purpose a simple case of introgression, the authors model a single pulse of admixture occurring recently (between the present and ¼ of the split time) with a fraction of individuals introgressed comprised between 10% and 50% (thus setting aside cases where introgression is rare, as expected around species barriers for example). This configuration is favourable for the detection of introgressed haplotypes, all the more so as population sizes are small, which faster the incomplete lineage sorting and reduces the risk of confounding it with introgression. We are convinced that testing the method against other scenarios, such as ancient migration or a scenario with continuous migration (not just admixture), would be more useful.

● We were surprised the authors used a forward-in-time simulator (SLiM) to simulate the first scenario, which is purely neutral as the other two scenarios. A coalescent approach seems more suited to this kind of simulation. By using SLiM, we would have expected the authors to explicitly include selection in their forward simulations (e.g. adaptive introgression or the presence of incompatible loci between species), which would have been useful to compare with their analysis of the Drosophila data. Another possibility to capture the effect of (linked) selection is to simulate under the coalescent and allow for heterogeneity in model parameters such as introgression rates or effective sizes (e.g. Roux et al. 2016 https://doi.org/10.1371/journal.pbio.2000234). This might not capture well the effect of adaptive introgression. Still, background selection and selection against migrants/hybrids can be well captured in that way, and these are also important widespread processes. In particular, it would be useful to assess how heterogeneity in incomplete lineage sorting across the genome (by varying Ne across loci in the simulations) affect the inference of introgressed tracts by IntroUNET. And more generally, it is important to delineate the conditions under which the method works correctly or not.

● The ghost introgression scenario is presented in Figure 3 as an ‘Archaic human ghost introgression scenario’. However, there are no biological justifications for the parameters chosen here. The authors refer to the study by Durvasula, which seems insufficient. Furthermore, the two methods (IntroUNET and ArchIE) require a reference non-introgressed population to test for ghost introgression in the focal population. However, it could be tough to know a priori which population is suitable as a reference in the case of ghost introgression because the source lineage is typically unknown. Methods without references are more appropriate in that situation (e.g. Chen et al. 2020 https://doi.org/10.1016/j.cell.2020.01.012). So the authors should clarify that IntroUNET is most suited for cases where the source population is suspected but unsampled or extinct (not unknown).

2. Applying to real data

● The authors clearly show that their method can identify introgressed alleles/haplotypes in simple demographic scenarios. However, when one goes to more complex (and so realistic) scenarios, IntroUNET seems to face some difficulties. Notably, the authors indicate along L334 that the Drosophila version of IntroUNET performs best when some introgression is present in the input. Therefore, they present results on regions previously found to contain introgression in the required direction (i.e. from D. simulans to D. sechellia).

The first issue is that most users of the method won’t know a priori the regions that introgress between their study species. Therefore, one should probably expect poorer evaluation results than those presented in Figure 6.

The second issue is that this version of IntroUNET occasionally produces many false positives. And the way false positives are handled seems not satisfactory. Indeed the authors simply discarded sites that were predicted to be introgressed in more than half of the recipient genomes, considering these are artefacts of their predictions. This may be a suitable method in situations where introgression events are known a priori, as in this Drosophila case. But it can never be a general solution as, in most situations, we won’t be able to disentangle artefacts from biological signals, i.e. the actual number of true positive targets isn’t typically known. This is an issue as the article aims to present a new general tool. These caveats on the applicability of the method should be spelt out.

● In the Drosophila scenario, the demographic parameters were first estimated with ∂a∂i under a model of isolation-with-migration and exponential population size change. However, the simulations were performed using single-pulse introgression, which isn’t justified. If the idea was to show the identification of the introgressed haplotypes is robust to deviations from the simulated scenario by IntroUNET, this should be highlighted.

● We think that Figure 7, which shows the distributions of introgressed allele frequencies, doesn’t really convey well the results the authors want to highlight. To help see the difference between categories (panel A: genic vs intergenic, panel B: sweep vs outside the sweep), the authors could draw vertical lines showing the average of each category (as reported in the text). Nevertheless, the differences in the average introgressed frequencies are quite small (3.2 vs 3.7 in the two comparisons), providing weak evidence that introgression is stronger in intergenic rather than genetic regions or stronger in the sweep region. It would be more informative, at least for the adaptive introgression case (panel B), to plot the length of the introgressed haplotypes as a function of their frequency, as long coincident haplotypes are expected around positively selected sites (as shown in Supp. Figure 5). As a note, it is a bit surprising that the adaptively introgressed alleles are not fixed in the recipient population: is that because the sweep is ongoing?

● The dimension of the input data in the simulations seems quite small (100 Kb regions analysed in windows of 125 polymorphisms) if the idea is to use IntroUNET to scan genomes for introgressed alleles/haplotypes. It would be helpful to specify in the 4.5 section the computational time (and under which multithread setting) when scanning for genomes of, say 500 Mb or 1 Gb.

● Another feature of the data that is not clearly spelt out is that haplotypes are required. This is important to specify early in the manuscript as potential users need to know which data type suits the method. Typically, it is still not common to get phased data (or have at its disposal inbred lines). So this limits the applicability of the IntroUNET method as it is presented and tested here (i.e. against haplotypes).

3. IntroUNET

● This deep learning algorithm for semantic segmentation allows classifying alleles into two classes: introgressed or non-introgressed. However, it is not designed to infer the best parameter values of the demographic scenario modelled, such as the time of admixture. We suggest the authors spell out this is not the aim of their method, as it could be confusing for readers unfamiliar with such a classification method.

● Some choices are not precisely justified. For example, it is unclear why the authors used SLiM in the first scenario (simple bidirectional introgression) while they used “msmodified” in the second (ghost introgression) and third (introgression in Drosophila) scenarios.

● The authors provide in Figure 3 the typical outputs of IntroUNET. But it is unclear how ones go from the matrix of the inferred introgression probabilities to the matrix of the inferred introgressed haplotypes (i.e. the binary classification). Did the authors apply an arbitrary probability threshold to classify each allele (e.g. if prob>0.5, then the “introgressed class” is depicted)?

● The authors mention along L210-2016 that the imbalance in the data (i.e. the fact the “introgressed” class is much less frequent than the “non-introgressed” class) may affect evaluation metrics. However, it is not clear from the text to what extent the configuration of the weighted form of cross-entropy affects the predictions made by the network, e.g. if the data is heavily imbalanced. Could you please explain more clearly what has been done to handle this issue?

● In the next paragraph (L217 to 226), the authors mention some technical constraints associated with the input image (the dimensions must be multiple of 16). Therefore, a step of individual up-sampling has to be applied. Again, the implication of such handling is not discussed in the manuscript.

● Nor discussed is the difference between IntroUNET and ArchIE regarding inputs: the first focuses on polymorphic sites, while the second includes invariants. Is that important to include invariants for the classification method, or does it just burden the simulations?

● Evaluation metrics are calculated globally, considering all genomic windows under study. However, ones can expect the ability of the method to identify introgressed haplotypes to depend on local features along the genome, e.g. local recombination rates. It would be useful to assess its effect by running a different version of the first scenario where the recombination rate is a magnitude lower than the mutation rate.

4. Technicalities

As a general note, specific technical aspects in the manuscript would not need to be presented in the article's main body considering the expected readers of this journal. We suggest adding a Supplementary Information section with the deep learning details (e.g. the learning rate, the actual loss function used, the potential issues of class imbalance and ordering individuals, etc.). This change will make room for important points that need to be explained in the current version (as the reader is currently expected to have some basics of deep learning, which won’t probably be the case).

● The « 3.1.1 Ordering individuals within the input image » section is presented a bit too early. We suggest moving it after the current part 3.1.2, which covers the network architecture, training and evaluation. We also suggest reducing it significantly in the main body and leaving the details for the Supp. Info. We acknowledge that resolving the seriation problem is very important when using this type of deep learning (and it illustrates that convolutional features are not perfectly suited for images such as genomic alignments). Still, it appears too detailed for the intended public of the journal (which, for the most part, will have a good background in genetics but no advanced knowledge in mathematics or computer sciences).

● The « 3.1.2 Network architecture, training, and evaluation » section lacks important information, in our opinion. Going over some basics of deep learning, especially regarding training and evaluation of the method, would be helpful for the most inexperienced readers. In addition, it would be difficult for a reader unfamiliar with deep learning methods to understand Figure 2: it is unclear what numbers under brackets are, why they change through the convolutional network, and why there are two matrices in the output. These points should be clarified in the legend. Moreover, an effort must be made to explain to non-experts the IntroUNet architecture described from L195 to L204. On L225, please, explain what a batch size is.

● Precision, recall, AUPR and AUROC curves are expected to be already known notions for the readers. It might be good to give short reminders as to what each of these metrics represents. Stating that these metrics evaluate the network's ability to make correct predictions without giving some reminders of what they mean is unsatisfying. For example, the PR curve for the “ghost introgression” scenario is worse than the one for the previous scenario, and the shape of the curve contains interesting information that the reader should understand.

● The use of Platt scaling seems to be helpful considering the improvement in accuracy (Figure 6), but once again, it is hard to understand this procedure without prior knowledge. We suggest adding information about Platt scaling in the Supp. Info. Furthermore, according to Supp. Figure 4, it seems that IntroUNET (before scaling) tends to consistently overestimate the introgression probability. Could you please comment on that?

5. Minor comments

● Section 3.2.1: as we understand, populations A and B are drawn from the same set of parameters. If that’s the case, what is the point of studying both the introgression from A to B and B to A (see Figure 4)? Is that for showing IntroUNET can correctly detect the introgression rate in each direction if it is asymmetric? Please, clarify this point. Moreover, it is unclear from the text that the fraction of individuals introgressed can vary freely in each direction (i.e. it can be asymmetric); please clarify this. Finally, in Table 1 (and also in Table 2) it would be more precise to replace “Migration time” with “Time of introgression event”, and “Migration probability” with “Fraction of individuals introgressed”.

● Sections 3.2.2 & 3.2.3: the authors do not specify the specific parameters used to train their network in IntroUNET. Please specify what the numbers of epochs are, the patience, etc.

● Section 3.2.3: Label smoothing is used in the Drosophila dataset to add some noise to the labels, and it seems to help with the training, as stated by the authors on L326. But this is a bit vague and would require further explanations.

● Section 4.3: on L416, it is unclear what are these “43 inferred demographic models” since a single scenario of isolation-with-migration was modelled with ∂a∂i. Please, clarify if they correspond to different multi-parameter estimates under the same demographic model. From L436 on, it would be more meaningful for readers if the numbers of introgressed haplotypes were transformed into fractions. Furthermore, the simulation setting (i.e. the recombination rate, the mutation rate, etc.) used in the Drosophila case is not stated in the methods.

● Section 4.5: this kind of comparison is useful, but it would be clearer to present it in a table. Moreover, some more « general guidelines » as to how long the execution of this pipeline would take, considering other concrete examples, would be a welcomed addition. The indication on L574 that using IntroUNET may involve experimentation to identify the optimal network architecture should perhaps also be included in the computational costs calculated in Section 4.5.

6. Typos & formatting

● L271: “as we described in 1”, please explain what “1” refers to.

● L343: typo in “recorded [the] the fraction (...)”.

● L343: “to have experienced introgressed at that site” seems a repeat of the start of the sentence.

● L367: “Reference-free inference of archaic local ancestry”: the title is confusing as the second scenario (archaic introgression) requires a target population and a reference un-introgressed population. Please, clarify.

● L376: typo in “Again, we trained our [ ] and tested our (…)”.

● L455: brackets are missing for “Supplementary Figure 5”.

● L572: typo in “a high-performance compute[] cluster (…)”.

● In the discussion, at several places, a space is missing after the species name D. simulans and D. sechellia.

● Legend of Table 1: “After some amount of time of complete isolation, which follows the described uniform distribution”. Isn’t the split time fixed in the simulations (ts=4N with N=500)?

● Legend of Figure 3: replacing “split followed by recent gene flow” with “split followed by a single-pulse introgression event” would be more precise.

● Legend of Figure 5: the legend does not indicate panels C and D.

● Supp. Figure 1: please, scale the introgression probability panels to that of the other panels.

● Supp. Figure 2: although it is stated that two examples without introgression are shown for each method, this seems not to be the case for Archie, as three examples have nearly zero introgressed haplotypes in the true panels.

● Legend of Supp. Figure 2: typo in “Each [ ] shows the true and (...)”.

● Supp. Figure 5: please, provide the unit of the x-axes.

Reviewer #3: See attached document

**Have all data underlying the figures and results presented in the manuscript been provided?**

Reviewer #1: Yes

Reviewer #2: Yes

Reviewer #3: Yes

PLOS authors have the option to publish the peer review history of their article (what does this mean?). If published, this will include your full peer review and any attached files.

Reviewer #1: No

Reviewer #2: No

Reviewer #3: No

---

## [Decision Letter · Decision Letter 1]

30 Nov 2023

Dear Dr Schrider,

Thank you very much for submitting your Research Article entitled 'IntroUNET: identifying introgressed alleles via semantic segmentation' to PLOS Genetics.

The manuscript was fully evaluated at the editorial level and by independent peer reviewers. The reviewers appreciated the attention to an important topic but identified some concerns that we ask you address in a revised manuscript.

We therefore ask you to modify the manuscript according to the review recommendations. Your revisions should address the specific points made by each reviewer.

Yours sincerely,

Nicolas Bierne

Academic Editor

PLOS Genetics

Kirsten Bomblies

%CORR_ED_EDITOR_ROLE%

PLOS Genetics

Dear Dr Schrider,

Thank you for your work of revision, which has taken into account the concerns of all the referees. They are all satisfied with the new version. I just let you deal with the minor revision requested by referee 3.

Best regards,

Nicolas Bierne

Reviewer's Responses to Questions

**Comments to the Authors:**

Reviewer #1: I would like to thank the authors for their thoughtful consideration of the reviewers’ comments and their extensive efforts to address all of our concerns. This was a lot of extra work, but well worth it! The improvements to the text have made the technical aspects of the project more accessible, and the additional testing of the method have provided meaningful new insight. In particular, I was really surprised to see an appreciable impact of background selection.

The revised manuscript is good and I do not have any further comments or concerns.

Best regards,

Derek Setter

Reviewer #2: The authors have made substantial efforts to address all of the major concerns raised by the reviewers. It is rare enough to merit mention, and I would like to thank the authors for their revisions. The updated version of the manuscript includes many new analyses and results to demonstrate the robustness of the method to various model misspecifications. They also extended the IntroUNET package to improve its accuracy with real genomic data (adding a first step of window-based classification) and they show its applicability to unphased data. Finally, the authors rendered the methods more accessible to non-experts and provided guidelines, which is an important aspect for the readership of Plos Genetics. A minor negative point is that the paper is quite long with many technical details in the main text. Overall, this work is of high quality and I expect that IntroUNET will be very useful to the community for genomic studies on introgression. Therefore, I fully support its publication.

Reviewer #3: This is a revised version of a manuscript I reviewed some time ago. My review included three main comments. The first one was focused on the preprocessing steps and how the choice of metrics/algorithms could influence performance of the method. The authors have now explored different distance metrics for the seriation step of the first population and identified those that work best. For the second step, they argue that the choice of the population for the initial step is not a major issue. I agree that with two populations, this may be the case, but it may be very different when more than two populations are considered. Nevertheless, the authors implementation considers a two-population scenario so this is not a major issue. However, the authors should not suggest that their method can easily be generalised to scenarios with more than two populations (c.f., ln. 147).

The second major comment was about the effects of model misspecification, which the authors have addressed in great detail in the present version. Finally, there was also an issue about the applicability of the method, because there was a substantial number of false positives when analysing regions that did not have introgression. To avoid this problem it was necessary to know in advance which regions were introgressed. To solve this issue, the authors now include a windows-based step that allows the identification of regions with introgression.

I only have two relatively minor comments that I would like the authors to address. They all concern the lack of a serious hyperparameter optimisation for the architecture used by the authors. In general, the optimisation step is considered as an important aspect of developing DNNs. Although the authors do acknowledge the absence of this step, they should not make statements that are not properly supported. This is particular the case for the explanation of label smoothing. It is unclear if the value of alpha chosen is high enough to have a regularisation effect. Typically, you need to do some experimentation in order to find the optimal value, but this was not done. Therefore, you cannot really state “We found this extra regularization was helpful for training". You need to present some evidence that you have really checked if it helps to avoid overfitting.

**Have all data underlying the figures and results presented in the manuscript been provided?**

Reviewer #1: Yes

Reviewer #2: Yes

Reviewer #3: Yes

PLOS authors have the option to publish the peer review history of their article (what does this mean?). If published, this will include your full peer review and any attached files.

Reviewer #1: **Yes: **Derek Setter

Reviewer #2: No

Reviewer #3: No

---

## [Editor Report · Decision Letter 2]

29 Jan 2024

Dear Dr Schrider,

We are pleased to inform you that your manuscript entitled "IntroUNET: identifying introgressed alleles via semantic segmentation" has been editorially accepted for publication in PLOS Genetics. Congratulations!

Yours sincerely,

Nicolas Bierne

Academic Editor

PLOS Genetics

Kirsten Bomblies

%CORR_ED_EDITOR_ROLE%

PLOS Genetics

Comments from the reviewers (if applicable):

Dear Dr Schrider,

Thank you for your final efforts to revise your manuscript. I am very pleased to now recommend acceptance of your manuscript for publication in PLoS Genetics.

Thank you for choosing PLoS Genetics to publish your very interesting method, which I am sure will be used by many of our readers.

Best regards,

Nicolas Bierne

**Data Deposition**

http://datadryad.org/submit?journalID=pgenetics&manu=PGENETICS-D-23-00137R2

**Press Queries**

---

## [Editor Report · Acceptance letter]

8 Feb 2024

PGENETICS-D-23-00137R2 

IntroUNET: identifying introgressed alleles via semantic segmentation 

Dear Dr Schrider, 

We are pleased to inform you that your manuscript entitled "IntroUNET: identifying introgressed alleles via semantic segmentation" has been formally accepted for publication in PLOS Genetics! Your manuscript is now with our production department and you will be notified of the publication date in due course.

With kind regards,

Anita Estes

PLOS Genetics

On behalf of:
